# RETHINKING OUT-OF-DISTRIBUTION DETECTION ON IMBALANCED DATA DISTRIBUTION

## ABSTRACT

Detecting and rejecting unknown out-of-distribution (OOD) samples is critical for deployed neural networks to void unreliable predictions. In real-world scenarios, however, the efficacy of existing OOD detection methods is often impeded by the inherent imbalance of in-distribution (ID) data, which causes significant performance decline. Through statistical observations, we have identified two common challenges faced by different OOD detectors: misidentifying tail class ID samples as OOD, while erroneously predicting OOD samples as head class from ID. To explain this phenomenon, we introduce a generalized statistical framework, termed ImOOD, to formulate the OOD detection problem on imbalanced data distribution. Consequently, the theoretical analysis reveals that there exists a class-aware *bias* item between balanced and imbalanced OOD detection, which contributes to the performance gap. Building upon this finding, we present a unified perspective of post-hoc normalization and training-time regularization techniques to calibrate and boost the imbalanced OOD detectors. On the representative CIFAR10-LT, CIFAR100-LT, and ImageNet-LT benchmarks, our method consistently surpasses the state-of-the-art OOD detection approaches by a large margin.

## 1 INTRODUCTION

Identifying and rejecting unknown samples during models' deployments, aka OOD detection, has garnered significant attention and witnessed promising advancements in recent years (Yang et al., 2021b; Bitterwolf et al., 2022; Ming et al., 2022; Tao et al., 2023a). Nevertheless, most advanced OOD detection methods are designed and evaluated in ideal settings with category-balanced in-distribution (ID) data. However, in practical scenarios, long-tailed class distribution (a typical imbalance problem) not only limits classifiers' capability (Buda et al., 2018), but also causes a substantial performance decline for OOD detectors (Wang et al., 2022).

Several efforts have been applied to enhance OOD detection on imbalanced data distributions (Wang et al., 2022; Huang et al., 2023; Sapkota & Yu, 2023). They mainly attribute the performance degradation to misidentifying samples from tail classes as OOD (due to the lack of training data), and concentrate on improving the discriminability for tail classes and out-of-distribution samples (Wang et al., 2022). Whereas, we argue that the confusion between tail class and OOD samples presents only one aspect of the imbalance problem arising from the long-tailed data distribution.

To comprehensively understand the imbalance issue, we employed several representative OOD detection methods (*i.e.*, OE (Hendrycks et al., 2019), Energy (Liu et al., 2020), and PASCL (Wang et al., 2022)) on the CIFAR10-LT dataset (Cao et al., 2019). For each model, we statistic the distribution of wrongly detected ID samples and wrongly detected OOD samples, respectively. The results in Fig. 1a reveal that different approaches encounter the same two challenges: (1) ID samples from tail classes are prone to be detected as OOD, and (2) OOD samples are prone to be predicted as ID from head classes. As illustrated in Fig. 1b, we argue that the disparate ID decision spaces on head and tail classes *jointly* result in the performance decline for OOD detection.

This paper introduces a generalized statistical framework, namely ImOOD, to formulate imbalanced OOD detection (see Section 3.1). Specifically, we extend closed-set ID classification to open-set scenarios and derive a unified posterior probabilistic model for ID/OOD identification. Consequently, we find that between balanced and imbalanced ID data distributions exists a class-aware *bias* item, which concurrently explains the inferior OOD detection performance on both head and tail classes.

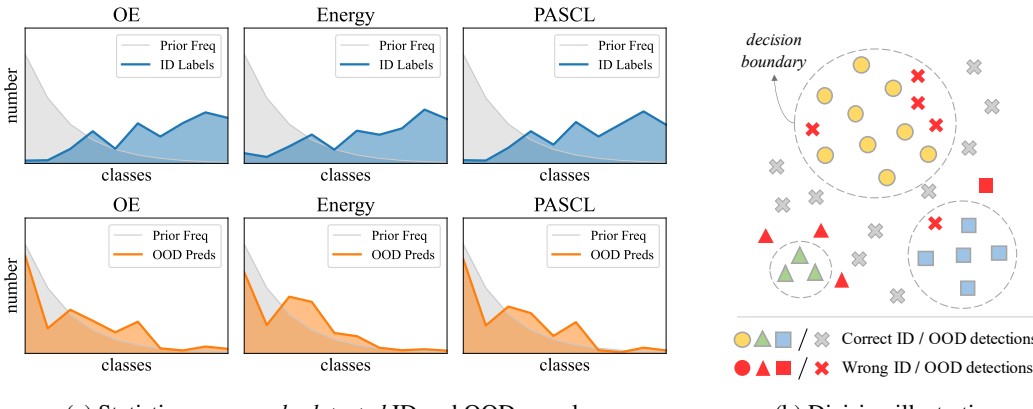

(a) Statistics on *wrongly-detected* ID and OOD samples.          (b) Dicision illustration.

Figure 1: **Issues of OOD detection on imbalanced data**. (a) Statistics of the class labels of ID samples that are wrongly detected as OOD, and the class predictions of OOD samples that are wrongly detected as ID. (b) Illustration of the OOD detection process in feature space. Head classes' huge decision space and tail classes' small decision space *jointly* damage the OOD detection.

Based on the generalized ImOOD framework, given an imbalanced OOD detector $g$, we develop a naive post-hoc normalization approach to adaptively calibrate the OOD detection probability via ID classification information (see Section 3.2). To further enhance the OOD detection on imbalanced distribution, we design a unified loss function to regularize the posterior ID/OOD probability during training (see Section 3.3), which simultaneously encourages the separability between tail ID classes and OOD samples, and prevents predicting OOD samples as head ID classes. Furthermore, ImOOD can readily generalize to various OOD detection methods, including OE (Hendrycks et al., 2019), Energy (Liu et al., 2020), and BinDisc (Bitterwolf et al., 2022), *etc.* With the support of theoretical analysis, our generalized statistical framework consistently translates into strong empirical performance on the CIFAR10-LT, CIFAR100-LT (Cao et al., 2019), and ImageNet-LT (Wang et al., 2022) benchmarks (see Section 4).

Our contribution can be summarized as follows:

- Through statistical observation and theoretical analysis, we reveal that OOD detection approaches collectively suffer from the disparate decision spaces between tail and head classes in the imbalanced data distribution.
- We establish a generalized statistical framework to formulate and explain the imbalanced OOD detection issue, and further provide a unified view of post-hoc normalization and loss modification techniques to alleviate the problem.
- We achieve superior OOD detection performance on three representative benchmarks, *e.g.*, outperforming the state-of-the-art method by 1.7% and 5.2% of AUROC on CIFAR10-LT and ImageNet-LT, respectively.

## 2 PRELIMINARIES

**Imbalanced Image Recognition**. Let $\mathcal{X}^{in}$ and $\mathcal{Y}^{in} = \{1, 2, \cdots, K\}$ denote the ID feature space and label space with $K$ categories in total. Let $\boldsymbol{x} \in \mathcal{X}^{in}$ and $y \in \mathcal{Y}^{in}$ be the random variables with respect to $\mathcal{X}^{in}$ and $\mathcal{Y}^{in}$. The posterior probability for predicting sample $\boldsymbol{x}$ into class $y$ is given by:

$$P(y|\boldsymbol{x}) = \frac{P(\boldsymbol{x}|y) \cdot P(y)}{P(\boldsymbol{x})} \propto P(\boldsymbol{x}|y) \cdot P(y) \tag{1}$$

Given a learned classifier $f \colon \mathcal{X}^{in} \mapsto \mathbb{R}^K$ that estimates $P(y|\boldsymbol{x})$, in the class-imbalance setting where the label prior $P(y)$ is highly skewed, $f$ is evaluated with the balanced error (BER) (Brodersen et al., 2010; Menon et al., 2013; 2021):

$$\text{BER}(f) = \frac{1}{K} \sum_y P_{\boldsymbol{x}|y}(y \neq \text{argmax}_{y'} f_{y'}(\boldsymbol{x})) \tag{2}$$

This can be seen as implicitly estimating a class-balanced posterior probability (Menon et al., 2021):

$$P^{\text{bal}}(y|\boldsymbol{x}) \propto \frac{1}{K} \cdot P(\boldsymbol{x}|y) \propto \frac{P(y|\boldsymbol{x})}{P(y)} \tag{3}$$

The ideal Bayesian-optimal classification becomes $y^* = \text{argmax}_{y \in [K]} P^{\text{bal}}(y|\boldsymbol{x})$.

**Out-of-distribution Detection**. In the open world, input sample $x$ may also come from the OOD feature space $\mathcal{X}^{out}$. Let $o$ be the random variable for an unknown label $o \notin \mathcal{Y}^{in}$, and $i$ be the union variable of all ID class labels (*i.e.*, $i = \cup y$). To ensure OOD detection is learnable (Fang et al., 2022), we make a mild assumption that the ID and OOD data space is separate (*i.e.*, $\mathcal{X}^{in} \cap \mathcal{X}^{out} = \emptyset$) and finite (*i.e.*, $|\mathcal{X}^{in}|, |\mathcal{X}^{out}| < \infty$ and $P(i), P(o) < 1$). Given an input $\boldsymbol{x}$ from the union space $\mathcal{X}^{in} \cup \mathcal{X}^{out} \triangleq \mathcal{X}$, the posterior probability for identifying $\boldsymbol{x}$ as in-distribution is formulated as:

$$P(i|\boldsymbol{x}) = \sum_y P(y|\boldsymbol{x}) = 1 - P(o|\boldsymbol{x}) \tag{4}$$

Correspondingly, $P(o|\boldsymbol{x})$ measures the probability that sample $\boldsymbol{x}$ does not belong to any known ID class, aka OOD probability. Hence, the OOD detection task can be viewed as a binary classification problem (Bitterwolf et al., 2022). Given a learned OOD detector $g \colon \mathcal{X}^{in} \cup \mathcal{X}^{out} \mapsto \mathbb{R}^1$ that estimate the $P(i|\boldsymbol{x})$, samples with lower scores $g(\boldsymbol{x})$ are detected as OOD and vice versa.

## 3 METHOD

### 3.1 BALANCED OOD DETECTION ON IMBALANCED DATA DISTRIBUTION

When ID classification meets OOD detection, slightly different from Eq. (1), the classifier $f$ is actually estimating the posterior class probability for sample $\boldsymbol{x}$ from ID space $\mathcal{X}^{in}$ merely, that is, $P(y|\boldsymbol{x}, i)$ (Hendrycks et al., 2019; Wang et al., 2022). Considering a sample $\boldsymbol{x}$ from the open space $\boldsymbol{x} \in \mathcal{X}^{in} \cup \mathcal{X}^{out}$, since $P(i, y|\boldsymbol{x}) = P(\cup y, y|\boldsymbol{x}) = P(y|x)$, Eq. (1) can be re-formulated as:

$$P(y|\boldsymbol{x}) = P(i, y|\boldsymbol{x}) = \frac{P(\boldsymbol{x}, i, y)}{P(\boldsymbol{x})} = \frac{P(\boldsymbol{x}, i, y)}{P(\boldsymbol{x}, i)} \cdot \frac{P(\boldsymbol{x}, i)}{P(\boldsymbol{x})} = P(y|\boldsymbol{x}, i) \cdot P(i|\boldsymbol{x}) \tag{5}$$

Recall that $P(y|\boldsymbol{x}, i)$ and $P(i|\boldsymbol{x})$ are estimated by $f$ and $g$, respectively. Combining Eq. (5) with Eq. (3), the balanced posterior class probability in the open-set is formulated as:

$$P^{\text{bal}}(y|\boldsymbol{x}) \propto \frac{P(y|\boldsymbol{x})}{P(y)} = \frac{P(y|\boldsymbol{x}, i)}{P(y)} \cdot P(i|\boldsymbol{x}) \tag{6}$$

Combining Eq. (6) with Eq. (4), the ideal balanced ID/OOD detection probability is given by:

$$P^{\text{bal}}(i|\boldsymbol{x}) = \sum_y P^{\text{bal}}(y|\boldsymbol{x}) \propto \sum_y \frac{P(y|\boldsymbol{x}, i)}{P(y)} \cdot P(i|\boldsymbol{x}) = P(i|\boldsymbol{x}) \cdot \sum_y \frac{P(y|\boldsymbol{x}, i)}{P(y)} \tag{7}$$

For simplicity, Eq. (7) is summarized as:

$$P^{\text{bal}}(i|\boldsymbol{x}) \propto P(i|\boldsymbol{x}) \cdot \sum_y \frac{P(y|\boldsymbol{x}, i)}{P(y)} \tag{8}$$

Compared to the original OOD probability $P(i|\boldsymbol{x})$, the *bias* item $\sum_y \frac{P(y|\boldsymbol{x},i)}{P(y)} \triangleq \beta(\boldsymbol{x})$ causes the performance gap for OOD detection on balanced and imbalanced data distribution.

**On class-balanced distribution**, the class-priority $P(y)$ is a constant that equals to each other of the ID categories, *i.e.*, $P(y) = const$. Since the summary of in-distribution classification probabilities equals to 1 (*i.e.*, $\sum_y P(y|\boldsymbol{x},i) = 1$), the bias item $\beta(\boldsymbol{x}) = \sum_y \frac{P(y|\boldsymbol{x},i)}{const} = \frac{1}{const} \sum_y P(y|\boldsymbol{x},i) = \frac{1}{const}$ is also a constant. Thus, according to Eq. (8), $P^{\text{bal}}(i|\boldsymbol{x}) \propto P(i|\boldsymbol{x})$, as $g(\boldsymbol{x})$ exactly models the balanced OOD detection.

**On class-imbalanced distribution**, the class-priority $P(y)$ is a class-specific value for ID categories. Since $\sum_y P(y|\boldsymbol{x},i) = 1$, the bias item $\beta(x) = \sum_y \frac{P(y|\boldsymbol{x},i)}{P(y)}$ can be viewed as a weighted sum of the reciprocal prior $\frac{1}{P(y)}$. According to Eq. (8), $\beta(x)$ causes the gap between balanced (ideal) ID/OOD probability $P^{\text{bal}}(i|\boldsymbol{x})$ and imbalanced (learned) $P(i|\boldsymbol{x})$, as $P^{\text{bal}}(i|\boldsymbol{x}) \propto P(i|\boldsymbol{x}) \cdot \beta(\boldsymbol{x})$:

- Given a sample $\boldsymbol{x}$ from an ID tail-class $y_t$ with a small prior $P(y_t)$, when the classification probability $P(y_t|\boldsymbol{x},i)$ gets higher, the term $\beta(\boldsymbol{x})$ becomes larger. Compared to the original $P(i|\boldsymbol{x})$ (learned by $g$), the calibrated $P(i|\boldsymbol{x}) \cdot \beta(\boldsymbol{x})$ is more likely to identify the sample $x$ as in-distribution, rather than OOD.
- Given a sample $\boldsymbol{x}'$ from OOD data, as the classifier $f$ tends to produce a higher head-class probability $P(y_h|\boldsymbol{x}',i)$ and a lower tail-class $P(y_t|\boldsymbol{x}',i)$ (Jiang et al., 2023), the term $\beta(\boldsymbol{x}')$ becomes smaller. Compared to the original $P(i|\boldsymbol{x}')$, the calibrated $P(i|\boldsymbol{x}') \cdot \beta(\boldsymbol{x}')$ is more likely to identify the sample $x'$ as out-of-distribution, rather than ID.

The above analysis is consistent with the statistical behaviors (see Fig. 1) of a vanilla OOD detector $g$. Compared to an ideal balanced detector $g^{\text{bal}}$, $g$ is prone to wrongly detect ID samples from tail class as OOD, and wrongly detect OOD samples as head class from ID.

In order to mitigate the imbalanced OOD detection, based on Eq. (8), this paper provides a unified view of post-hot normalization and training-time regularization techniques in the following sections.

## 3.2    Post-hoc Normalization

Given a vanilla OOD detector $g$ learned on imbalanced data distribution, Eq. (8) suggests one may use the posterior classification probability $P(y|\boldsymbol{x},i)$ and prior $P(y)$ to adaptively calibrate $P(i|\boldsymbol{x})$ to get close to the balanced $P^{\text{bal}}(i|\boldsymbol{x})$. We now show how to perform the naive post-hoc normalization from a statistical perspective.

**(1)** $P(i|\boldsymbol{x})$. Since OOD detection is a binary classification task (Bitterwolf et al., 2022), for an arbitrary OOD detector $g$ (Hendrycks et al., 2019; Liu et al., 2020), the posterior probability can be ultimately estimated by a *sigmoid* function $P(i|\boldsymbol{x}) := \frac{e^{g(\boldsymbol{x})}}{1+e^{g(\boldsymbol{x})}}$, where $g(\boldsymbol{x})$ is the ID/OOD logit.

**(2)** $P(y|\boldsymbol{x},i)$. Given a learned classifier $f$, the classification probability can be estimated by a *softmax* function $P(y|\boldsymbol{x},i) := \frac{e^{f_y(\boldsymbol{x})}}{\sum_{y'} e^{f_{y'}(\boldsymbol{x})}} \triangleq p_{y|\boldsymbol{x},i}$, where $f_y(\boldsymbol{x})$ presents the logit for class $y$.

**(3)** $P(y)$. An empirical estimation of the prior $P(y)$ is using the label frequency of the training dataset (Cao et al., 2019; Menon et al., 2021). In the open-set, it becomes $P(y) := \frac{n_y}{n_o + \sum_{y'} n_{y'}} \triangleq \hat{\pi}_y$, where $n_o$ is the number of auxiliary OOD training samples, and $n_y$ indicates the number of class $y$.

For numerical stability (see Appendix A.3), we take the logarithm of both sides of Eq. (8), and the balanced OOD detector $g^{\text{bal}}$ is derived as:

$$g^{\text{bal}}(\boldsymbol{x}) := g(\boldsymbol{x}) + \log \sum_y \frac{p_{y|\boldsymbol{x},i}}{\hat{\pi}_y} \tag{9}$$

By adding the distribution-dependent offset $\log \sum_y \frac{p_{y|\boldsymbol{x},i}}{\hat{\pi}_y}$, $g^{\text{bal}}$ is able to bridge the gap between balanced and imbalanced data distributions. As analyzed in Section 3.1, $g^{\text{bal}}$ produces a higher ID probability for the sample $\boldsymbol{x}$ from ID tail classes, and generates a lower ID probability (*i.e.*, a higher OOD probability) for an OOD sample $\boldsymbol{x}'$ predicted as ID head classes.

To directly strengthen the OOD detection capability during training, we introduce the way to regularize the loss function in the following Section 3.3.

### 3.3 Training-time Regularization

According to Eq. (8), a more effective approach to balanced OOD detection is directly modeling the balanced posterior probability $P^{\text{bal}}(i|\boldsymbol{x})$, rather than the vanilla $P(i|\boldsymbol{x})$. To do so, consider the following regularized ID/OOD binary cross-entropy loss function:

$$\ell(g(\boldsymbol{x}), t) = \mathcal{L}_{\text{BCE}}\left(g(\boldsymbol{x}) - \log \sum_y \frac{p_{y|\boldsymbol{x},i}}{\pi_y}, \ t\right) \tag{10}$$

where $t = \mathbf{1}\{\boldsymbol{x} \in \mathcal{X}^{in}\}$ indicates whether the sample $\boldsymbol{x}$ comes from ID or not. Given a detector $g^*$ that minimizes the above Eq. (10), it has already estimated the balanced $P^{\text{bal}}(i|\boldsymbol{x})$. We thus predict the ID/OOD probability as usual: $\hat{p}(i|\boldsymbol{x}) = \frac{1}{1+e^{-g^*(\boldsymbol{x})}}$.

Note that in Eq. (10), we did not use the empirical statistics $\hat{\pi}_y = \frac{n_y}{n_o + \sum_{y'} n_{y'}}$ to regularize detector $g$. The reasons are two-fold: (1) $\hat{\pi}_y$ is sensitive to $n_o$, which means the choice of auxiliary OOD training set will change the prior estimates, while collecting an optimal OOD training set is generally prohibitive; (2) the empirical frequencies of $\hat{\pi}_y$ are not precise estimations for prior probabilities of $P(y)$ (Cao et al., 2019; Menon et al., 2021). Therefore, we estimate $P(y)$ by a set of learnable and input-agnostic parameters $\pi_y(\theta) \triangleq \pi_y$ in Eq. (10) via automatic gradient back-propagation during training. This idea is motivated by the recent prompt tuning frameworks (Lester et al., 2021; Jia et al., 2022; Zhou et al., 2022), which also aim to learn some prior properties from the joint distribution of $\mathcal{X} \times \mathcal{Y}$. The learnable $\pi_y$ is initialized with the statistical $\hat{\pi}_y$ to ensure stable convergence.

The ultimate framework, termed ImOOD, is verified through extensive experiments in Section 4.

## 4 Experiments

In this section, we empirically validate the effectiveness of our ImOOD on several representative imbalanced OOD detection benchmarks. The experimental setup is described in Section 4.1, based on which extensive experiments and discussions are displayed in Section 4.2 and Section 4.3.

### 4.1 Setup

**Datasets.** Following the literature (Wang et al., 2022; Jiang et al., 2023; Choi et al., 2023), we use the popular CIFAR10-LT, CIFAR100-LT (Cao et al., 2019), and ImageNet-LT (Liu et al., 2019) as imbalanced in-distribution datasets.

For CIFAR10/100-LT benchmarks, training data's imbalance ratio (*i.e.*, $\rho = \max_y(n_y)/\min_y(n_y)$) is set as 100, following Cao et al. (2019); Wang et al. (2022). The original CIAFR10/100 test sets are kept for evaluating the ID classification capability. For OOD detection, the TinyImages80M (Torralba et al., 2008) is adopted as the auxiliary OOD training data, and the test set is semantically coherent out-of-distribution (SC-OOD) benchmark (Yang et al., 2021a).

For the large-scale ImageNet-LT benchmark, we follow Liu et al. (2019); Wang et al. (2022) to select training samples from the whole ImageNet-1k (Deng et al., 2009) dataset, and take the original validation set for evaluation. We take the same OOD detection setting as Wang et al. (2022), where the ImageNet-Extra is used as auxiliary OOD training data and ImageNet-1k-OOD for testing. Randomly sampled from ImageNet-22k (Deng et al., 2009), ImageNet-Extra contains 517,711 images belonging to 500 classes, and ImageNet-1k-OOD consists of 50,000 images from 1,000 classes. All the classes in ImageNet-LT, ImageNet-Extra, and ImageNet-1k-OOD are not overlapped.

**Evaluation Metrics.** For OOD detection, we follow Hendrycks et al. (2019); Yang et al. (2021a); Wang et al. (2022) to report three metrics: (1) AUROC, the area under the receiver operating characteristic curve, (2) AUPR, the area under precision-recall curve., and (3) FPR95, the false positive rate of OOD samples when the true positive rate of ID samples are 95%. For ID classification,

we measure the macro accuracy of the classifier, as in Eq. (2). We report the mean and standard deviation of performance (%) over six random runs for each method.

**Implementation Details.** For the ID classifier $f$, following the settings of Wang et al. (2022), we train ResNet18 (He et al., 2016) models on the CIFAR10/100LT benchmarks, and leverage ResNet50 models for the ImageNet-LT benchmark. Logit adjustment loss (Menon et al., 2021) is adopted to alleviate the imbalanced ID classification. Detailed settings are displayed in Appendix A.1. For the OOD detector $g$, as Bitterwolf et al. (2022) suggest, we implement $g$ as a binary discriminator (abbreviated as *BinDisc*) to perform ID/OOD identification with Eq. (9) and Eq. (10). Detector $g$ shares the same backbone (feature extractor) as classifier $f$, and $g$ only attaches an additional output node to the classification layer of $f$. To verify the versatility of our method, we also implement several representative OOD detection methods (*e.g.*, OE (Hendrycks et al., 2019), Energy (Liu et al., 2020), *etc.*) into binary discriminators, and equip them with our ImOOD framework. For more details please refer to Appendix A.2.

**Methods for comparison.** In the following sections, we mainly compare our method with the recent OOD detectors including OE (Hendrycks et al., 2019), Energy (Liu et al., 2020), BinDisc (Bitterwolf et al., 2022), *etc.* Specifically, we conduct comprehensive comparisons with the advanced baseline PASCL (Wang et al., 2022), and also report the results of standard MSP (Hendrycks & Gimpel, 2017) and the modern variant PR+MSP (Jiang et al., 2023) scores on pre-trained models. In addition, Choi et al. (2023) have recently achieved superior performance on CIFAR10/100-LT benchmarks, whereas both of the training and testing procedures are different from PASCL's and our settings. We thus separately take their settings to assess our method in Appendix B.3 for a fair comparison.

## 4.2 MAIN RESULTS

Table 1: Post-hoc normalization for different OOD detectors on CIFAR10-LT.

| Method | AUROC (↑) | AUPR (↑) | FPR95 (↓) | ID ACC (↑) |
|---|---|---|---|---|
| OE | $89.77 \pm 0.06$ | $87.29 \pm 0.49$ | $\mathbf{33.17} \pm 0.93$ | $73.77 \pm 0.37$ |
| **+ImOOD** | $\mathbf{90.06} \pm 0.13$ | $\mathbf{88.36} \pm 0.30$ | $34.27 \pm 0.62$ | $73.77 \pm 0.37$ |
| Energy | $86.42 \pm 0.21$ | $80.64 \pm 0.41$ | $36.45 \pm 0.59$ | $74.67 \pm 0.51$ |
| **+ImOOD** | $\mathbf{87.68} \pm 0.17$ | $\mathbf{83.27} \pm 0.27$ | $\mathbf{35.13} \pm 0.48$ | $74.67 \pm 0.51$ |
| BinDisc | $87.37 \pm 0.23$ | $81.05 \pm 0.64$ | $33.66 \pm 0.12$ | $74.53 \pm 0.43$ |
| **+ImOOD** | $\mathbf{87.64} \pm 0.30$ | $\mathbf{81.61} \pm 0.63$ | $\mathbf{31.31} \pm 0.08$ | $74.53 \pm 0.43$ |

**ImOOD consistently boosts existing OOD detectors via post-hoc normalization.** To validate the efficacy of our generalized statistical framework, we first train several standard OOD detectors on the CIFAR10-LT (Cao et al., 2019) benchmark using popular methods like OE (Hendrycks et al., 2019), Energy (Liu et al., 2020), and BinDisc (Bitterwolf et al., 2022). Then we implement our ImOOD as a post-hoc normalization via Eq. (9) to adaptively adjust those OOD detectors' predictions with imbalanced classification information. The results are displayed in Table 1, which suggests our ImOOD brings consistent performance improvements for existing OOD detectors (*e.g.*, 0.3% increase on AUROC). Specifically, our ImOOD decreases 1.3% of FPR95 for Energy and 2.3% for BinDisc, and the post-hoc normalization does not affect ID classifications.

Though Table 1 indicates our ImOOD is effective in naively· normalizing the imbalanced OOD detectors, the final performance (*e.g.*, 90% of AUROC and 31% of FPR95) is still far from satisfaction for safe deployments. Hence, we employ our ImOOD as the training-time regularization via Eq. (10) to directly learn a balanced BinDisc detector on CIFAR10/100-LT and ImageNet-LT benchmarks (Wang et al., 2022), and the results are illustrated as follows.

**ImOOD significantly outperforms previous SOTA methods on CIFAR10/100-LT benchmarks.** As shown in Table 2, our ImOOD achieves new SOTA performance on both of CIFAR10/100-LT benchmarks, and surpasses the advanced PASCL (Wang et al., 2022) by a large margin. Specifically, ImOOD leads to 1.7% increase of AUROC, 2.1% increase of AUPR, and 5.1% decrease of FPR95 on CIFAR10-LT, with 0.8% - 1.7% enhancements of respective evaluation metrics on CIFAR100-LT. The detailed comparison with PASCL on six OOD test sets is reported in Appendix B.2.

Table 2: Performance on CIFAR10/100-LT benchmarks.

| Method | AUROC (↑) | AUPR (↑) | FPR95 (↓) | ID ACC (↑) |
|---|---|---|---|---|
| | ID Dataset: CIFAR10-LT | | | |
| MSP | $72.09 \pm 0.74$ | $70.40 \pm 0.44$ | $68.79 \pm 2.77$ | $72.75 \pm 0.12$ |
| RP+MSP | $72.42 \pm 0.57$ | $70.54 \pm 0.37$ | $69.06 \pm 1.57$ | $72.75 \pm 0.12$ |
| Energy | $86.42 \pm 0.21$ | $80.64 \pm 0.41$ | $36.45 \pm 0.59$ | $74.67 \pm 0.51$ |
| OE | $89.77 \pm 0.06$ | $87.29 \pm 0.49$ | $33.17 \pm 0.93$ | $73.77 \pm 0.37$ |
| PASCL | $90.99 \pm 0.19$ | $89.24 \pm 0.34$ | $33.36 \pm 0.79$ | $77.08 \pm 1.01$ |
| **ImOOD** (ours) | $\mathbf{92.73} \pm 0.33$ | $\mathbf{92.31} \pm 0.12$ | $\mathbf{28.27} \pm 0.32$ | $\mathbf{78.01} \pm 0.36$ |
| | ID Dataset: CIFAR100-LT | | | |
| MSP | $62.17 \pm 0.91$ | $57.99 \pm 0.45$ | $84.14 \pm 0.57$ | $41.01 \pm 0.31$ |
| RP+MSP | $62.43 \pm 0.68$ | $58.11 \pm 0.39$ | $83.97 \pm 0.44$ | $41.01 \pm 0.31$ |
| Energy | $69.77 \pm 0.57$ | $66.92 \pm 0.44$ | $75.37 \pm 0.65$ | $39.93 \pm 0.17$ |
| OE | $72.71 \pm 0.41$ | $67.01 \pm 0.21$ | $68.03 \pm 0.42$ | $39.24 \pm 0.34$ |
| PASCL | $73.32 \pm 0.32$ | $67.18 \pm 0.10$ | $67.44 \pm 0.58$ | $43.10 \pm 0.47$ |
| **ImOOD** (ours) | $\mathbf{74.14} \pm 0.23$ | $\mathbf{68.43} \pm 0.53$ | $\mathbf{65.73} \pm 0.16$ | $\mathbf{43.62} \pm 0.26$ |

The results suggest that training-time regularization is more effective in alleviating the imbalanced OOD detection problem. To further demonstrate the efficacy, we validate our method on the real-world large-scale ImageNet-LT (Liu et al., 2019) benchmark, and the results are displayed below.

**ImOOD achieves superior performance on the ImageNet-LT benchmark.** As Table 3 implies, our ImOOD brings significant improvements against PASCL, *e.g.*, 5.2% increase on AUROC and 12.7% decrease on FPR95. Since the performance enhancement is much greater than those on CIFAR10/100-LT benchmarks, we further statistic the class-aware error distribution on wrongly detected ID/OOD sample in Fig. A1. The results indicate our method builds a relatively better-balanced OOD detector on ImageNet-LT, which leads to higher performance improvements. Besides, as we employ ResNet18 on CIFAR10/100-LT while adopt ResNet50 on ImageNet-LT, the model capacity also seems to play a vital role in balancing the OOD detection on imbalanced data distribution, particularly in more challenging real-world scenarios.

Table 3: Performance on the ImageNet-LT benchmark.

| Method | AUROC (↑) | AUPR (↑) | FPR95 (↓) | ID ACC (↑) |
|---|---|---|---|---|
| MSP | $54.41 \pm 0.39$ | $51.25 \pm 0.45$ | $90.11 \pm 0.52$ | $41.05 \pm 0.67$ |
| RP+MSP | $54.63 \pm 0.25$ | $51.30 \pm 0.43$ | $90.34 \pm 0.46$ | $41.05 \pm 0.67$ |
| Energy | $63.54 \pm 0.34$ | $63.97 \pm 0.32$ | $88.52 \pm 0.20$ | $40.25 \pm 0.53$ |
| OE | $67.12 \pm 0.20$ | $69.31 \pm 0.26$ | $87.62 \pm 0.27$ | $39.27 \pm 0.32$ |
| PASCL | $68.11 \pm 0.28$ | $70.15 \pm 0.29$ | $87.37 \pm 0.41$ | $45.25 \pm 0.49$ |
| **ImOOD** (ours) | $\mathbf{73.31} \pm 0.30$ | $\mathbf{70.75} \pm 0.21$ | $\mathbf{74.67} \pm 0.45$ | $\mathbf{46.10} \pm 0.26$ |

In addition, our ImOOD brings better ID classification accuracy on all three imbalanced benchmarks. We owe it to the relatively disentangled optimization of ID classifier $f$ and OOD detector $g$, which is incidentally discussed in the last part of Section 4.3.

## 4.3 ABLATION STUDIES

In this section, we conduct in-depth ablation studies on the CIFAR10-LT benchmark to assess the validity and versatility of our proposed ImOOD framework, and the results are reported as follows.

**Automatically learning the class-prior $P(y) \triangleq \pi_y$ is effective.** To validate our training-time regularization in Eq. (10), we first build a baseline model with BinDisc (Bitterwolf et al., 2022) only, and no extra regularization is adopted to mitigate the imbalanced OOD detection. As shown in the first row from Table 4, the baseline (denoted as *none* of $P(y)$ estimates) presents a fair OOD

detection performance (*e.g.*, 90.06% of AUROC and 33.39% of FPR95). Then, we leverage the statistical estimates of label frequency (denoted as $\pi_y = \hat{\pi}_y$) to perform the loss regularization, and the performance receives an immediate 1.7%-2.3% improvement of all the AUROC, AUPR, FPR95 measures. Furthermore, automatically learning the class-prior (denoted as $\pi_y = \pi_y(\theta)$) leads to a consecutively better OOD performance (*e.g.*, 92.19% of AUROC and 29.29% of FPR95). The learned class-prior is visualized in Fig. 2, which indicates automatic prior estimation from existing data distribution is practical and effective.

Table 4: Estimates on the class-prior $P(y) \triangleq \pi_y$ for training-time regularization.

| $P(y)$ **Estimates** | **AUROC** ($\uparrow$) | **AUPR** ($\uparrow$) | **FPR95** ($\downarrow$) | **ID ACC** ($\uparrow$) |
|---|---|---|---|---|
| none | $90.06 \pm 0.48$ | $88.72 \pm 0.48$ | $33.39 \pm 0.25$ | $\mathbf{78.22} \pm 0.21$ |
| $\pi_y = \hat{\pi}_y$ | $91.79 \pm 0.37$ | $90.91 \pm 0.31$ | $31.03 \pm 0.21$ | $77.96 \pm 0.25$ |
| $\pi_y = \pi_y(\theta)$ | $\mathbf{92.19} \pm 0.33$ | $\mathbf{91.56} \pm 0.25$ | $\mathbf{29.29} \pm 0.49$ | $78.09 \pm 0.37$ |

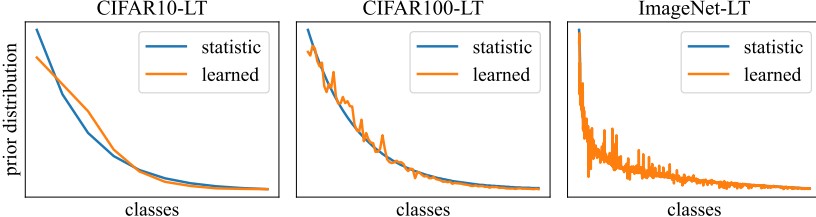

Figure 2: Visualization of the statistical label-frequency $\hat{\pi}_y$ and learned label-prior $\pi_y(\theta)$.

**ImOOD generalizes to various OOD detection methods.** To verify the versatility of our statistical framework, we implement the ImOOD framework with different OOD detectors, including OE (Hendrycks et al., 2019), Energy (Liu et al., 2020), and BinDisc (Bitterwolf et al., 2022). For OE and Energy detectors, we append an extra linear layer to conduct logistic regression on their vanilla OOD scores (*e.g.*, the maximum softmax probability for OE) (see Appendix A.2), which are thus universally formulated as the detector $g(\boldsymbol{x})$ in Eq. (10). Then, our training-time regularization is leveraged to optimize those detectors (denoted as **+ImOOD**). According to the results presented in Table 5, our ImOOD consistently boosts the original OOD detectors with stronger performance on all the AUROC, AUPR, and FPR95 measures. In particular, after utilizing the feature-level contrastive learning technique from PASCL (Wang et al., 2022) (denoted as **+ImOOD+PASCL**), our method achieves a superior OOD detection performance (*e.g.*, 92.73% of AUROC and 28.27% of FPR95). It implies our ImOOD framework does not conflict with current feature-level optimization approaches, and the combination of them may lead to better results. In Fig. A2, we also statistic the class distribution on wrongly-detected ID/OOD samples, which suggests our method continually mitigates the imbalanced problem for those original detectors. In real-world applications, one may choose a proper formulation of our ImOOD to meet the specialized needs.

Table 5: Training-time regularization on different OOD detectors.

| **Method** | **AUROC** ($\uparrow$) | **AUPR** ($\uparrow$) | **FPR95** ($\downarrow$) | **ID ACC** ($\uparrow$) |
|---|---|---|---|---|
| OE | $89.91 \pm 0.18$ | $87.32 \pm 0.24$ | $34.06 \pm 0.53$ | $75.75 \pm 0.38$ |
| **+ImOOD** | $\mathbf{91.66} \pm 0.29$ | $\mathbf{91.31} \pm 0.38$ | $\mathbf{32.81} \pm 0.40$ | $\mathbf{76.05} \pm 0.80$ |
| Energy | $90.27 \pm 0.16$ | $88.73 \pm 0.28$ | $34.42 \pm 0.57$ | $\mathbf{77.92} \pm 0.18$ |
| **+ImOOD** | $\mathbf{91.50} \pm 0.29$ | $\mathbf{90.52} \pm 0.24$ | $\mathbf{31.68} \pm 0.22$ | $77.75 \pm 0.39$ |
| BinDisc | $90.06 \pm 0.48$ | $88.72 \pm 0.48$ | $33.39 \pm 0.25$ | $\mathbf{78.22} \pm 0.21$ |
| **+ImOOD** | $92.19 \pm 0.33$ | $91.56 \pm 0.25$ | $29.29 \pm 0.49$ | $78.09 \pm 0.37$ |
| **+ImOOD+PASCL** | $\mathbf{92.73} \pm 0.33$ | $\mathbf{92.31} \pm 0.12$ | $\mathbf{28.27} \pm 0.32$ | $78.01 \pm 0.36$ |

In addition, as OE adopts the softmax output from ID classifier $f$ to perform OOD scorer $g$, the deeply coupled optimization on $f$ and $g$ may sacrifice the ID classification accuracy (Wang et al.,

2022; Choi et al., 2023). Meanwhile, Energy utilizes the logits (before softmax function) to perform OOD detection, whose ID accuracy obtains an increase (from 75.75% to 77.92%). As for BinDisc that only shares the features for $f$ and $g$, the relatively disentangled optimization leads to a higher classification accuracy (78.22%).

# 5 RELATED WORKS

**Out-of-distribution detection.** To reduce the overconfidence on unseen OOD samples (Bendale & Boult, 2015), a surge of post-hoc scoring functions has been devised based on various information, including output confidence (Hendrycks & Gimpel, 2017; Liang et al., 2018; Liu et al., 2023), free energy (Liu et al., 2020; Du et al., 2022; Lafon et al., 2023), Bayesian inference (Maddox et al., 2019; Cao & Zhang, 2022), gradient information (Huang et al., 2021), model/data sparsity (Sun et al., 2021; Zhu et al., 2022; Djurisic et al., 2023; Ahn et al., 2023), and visual distance (Sun et al., 2022; Tao et al., 2023b), *etc.* And the vision-language models like CLIP (Radford et al., 2021) have been recently leveraged to explicitly collect potential OOD labels (Fort et al., 2021; Esmaeilpour et al., 2022) or conduct zero-shot OOD detections (Ming et al., 2022). Another promising approach is adding open-set regularization in the training time (Malinin & Gales, 2018; Hendrycks et al., 2019; Wei et al., 2022; Wang et al., 2023b; Lu et al., 2023), making models produce lower confidence or higher energy on OOD data. Manually-collected (Hendrycks et al., 2019; Wang et al., 2023a) or synthesized outliers (Du et al., 2022; Tao et al., 2023b) are required for auxiliary constraints.

Works insofar have mostly focused on the ideal setting with balanced data distribution for optimization and evaluation. This paper aims at OOD detection on practically imbalanced data distribution.

**OOD detection on imbalanced data distribution.** In real-world scenarios, the deployed data frequently exhibits long-tailed distribution, and Liu et al. (2019) start to study the open-set classification on class-imbalanced setup. Wang et al. (2022) systematically investigate the performance degradation for OOD detection on imbalanced data distribution, and develop a partial and asymmetric contrastive learning (PASCL) technique to tackle this problem. Consequently, Jiang et al. (2023) propose to adopt class prior as a post-hoc normalization for pre-trained models, and Sapkota & Yu (2023) employ adaptive distributively robust optimization (DRO) to quantify the sample uncertainty from imbalanced distributions. Choi et al. (2023) focus on the imbalance problem in OOD data, and develop an adaptive regularization item for each OOD sample during networks' optimization.

Different from previous efforts, this paper establishes a generalized probabilistic framework to formularize the imbalanced OOD detection issue, and provides a unified view of post-hoc normalization and training-time regularization techniques to alleviate this problem.

# 6 CONCLUSION

This paper establishes a statistical framework, termed ImOOD, to formularize OOD detection on imbalanced data distribution. Through theoretical analysis, we find there exists a class-aware biased item between balanced and imbalanced OOD detection models. Based on it, our ImOOD provides a unified view of post-hoc normalization and training-time regularization techniques to alleviate the imbalance problem. On three popular imbalanced OOD detection benchmarks, extensive experiments and ablation studies are conducted to demonstrate the validity and versatility of our method. We hope our work can inspire new research in this community.

**Limitations and future work.** To ensure satisfactory OOD detection performance, ImOOD requires full-data learning, where auxiliary OOD training samples are needed. It has two major limitations: (1) the training cost is inevitable when the deployed environment changes, and fast adaptation across scenarios should be explored. (2) the manually collected auxiliary OOD data is relatively expensive, and recent outlier synthesis techniques may reduce the burden. We view those as our future work.

**Reproducibility statement.** This work follows the settings and code as PASCL (Wang et al., 2022) to implement our method. For reproducibility, necessary information has been provided in Section 4.1 and Appendix A in detail, involving adopted datasets, training and evaluation settings, implementation details, *etc.*

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

# Appendices

## A EXPERIMENTAL SETTINGS AND IMPLEMENTATIONS

### A.1 TRAINING SETTINGS

For a fair comparison, we mainly follow PASCL's (Wang et al., 2022) settings. On CIFAR10/100-LT benchmarks, we train ResNet18 (He et al., 2016) for 200 epochs using Adam optimizer, with a batch size of 256. The initial learning rate is 0.001, which is decayed to 0 using a consine annealing scheduler. The weight decay is $5 \times 10^{-4}$. On ImageNet-LT benchmark, we train ResNet50 (He et al., 2016) for 100 epochs with SGD optimizer with the momentum of 0.9. The batch size is 256. The initial learning rate is 0.1, which are decayed by a factor of 10 at epoch 60 and 80. The weight decay is $5 \times 10^{-5}$. During training, each batch contains an equal number of ID and OOD data samples (*i.e.*, 256 ID samples and 256 OOD samples).

For a better performance, one may carefully tune the hyper-parameters to train the models.

### A.2 IMPLEMENTATION DETAILS

In the manuscript, we proposed a generalized statistical framework to formularize and alleviate the imbalanced OOD detection problem. This section provides more details on implementing different OOD detectors into a unified formulation, *e.g.*, a binary ID/OOD classifier Bitterwolf et al. (2022):

- For BinDisc (Bitterwolf et al., 2022), we simply append an extra ID/OOD output node to the classifier layer of a standard ResNet model, where ID classifier $f$ and OOD detector $g$ share the same feature extractor. Then we adopt the sigmoid function to convert the logit $g(\boldsymbol{x})$ into the ID/OOD probability $\hat{p}(i|\boldsymbol{x}) = \frac{1}{1+e^{-g(\boldsymbol{x})}}$.

- For Energy (Liu et al., 2020), following Du et al. (2022), we first compute the negative free-energy $E(\boldsymbol{x}; f) = \log \sum_y e^{f_y(\boldsymbol{x})}$, and then attach an extra linear layer to calculate the ID/OOD logit $g(\boldsymbol{x}) = w \cdot E(\boldsymbol{x}; f) + b$, where $w, b$ are learnable scalars. Hence, the sigmoid function is able to convert the logit $g(\boldsymbol{x})$ into the probability $\hat{p}(i|\boldsymbol{x})$.

- For OE (Hendrycks et al., 2019), similarly, we compute the maximum softmax-probability $\mathrm{MSP}(\boldsymbol{x}; f) = \max_y \frac{e^{f_y(\boldsymbol{x})}}{\sum_{y'} e^{f_{y'}(\boldsymbol{x})}}$, and use another linear layer to obtain the ID/OOD logit $g(\boldsymbol{x}) = w \cdot \mathrm{MSP}(\boldsymbol{x}; f) + b$.

By doing so, one may leverage Eq. (9) as the post-hoc normalization for a given model, or exploit Eq. (10) as the training-time regularization to derive a strong OOD detector.

### A.3 DISCUSSION ON REGULARIZATION IMPLEMENTATION

In Section 3.1, we model the balanced ID/OOD probability as $P^{\mathrm{bal}}(i|\boldsymbol{x}) \propto P(i|\boldsymbol{x}) \cdot \sum_y \frac{P(y|\boldsymbol{x},i)}{P(y)}$, and respectively use the softmax-output $p_{y|\boldsymbol{x},i}$ and label-frequency $\hat{\pi}_y$ to estimate $P(y|\boldsymbol{x},i)$ and $P(y)$. An intuitive quantification of $P^{\mathrm{bal}}(i|\boldsymbol{x})$ is $P^{\mathrm{bal}}(i|\boldsymbol{x}) \propto P(i|\boldsymbol{x}) \cdot \sum_y \frac{p_{y|\boldsymbol{x},i}}{\hat{\pi}_y} \triangleq P(i|\boldsymbol{x}) \cdot \hat{\beta}(\boldsymbol{x})$, which however may cause numerical instability. Taking CIFAR10-LT (Cao et al., 2019) with imbalance ratio $\rho = 100$ as an example, the reciprocal priority for tail-class $y_t$ becomes $\frac{1}{\hat{\pi}_{y_t}} \approx 248$. Even if the classifier $f$ produces a relatively lower posteriority (*e.g.*, $p_{y_t|\boldsymbol{x},i} = 0.1$), the bias item becomes $\hat{\beta}(\boldsymbol{x}) \triangleq \sum_y \frac{p_{y|\boldsymbol{x},i}}{\hat{\pi}_y} > 0.1 \times 248 = 24.8$, which may overwhelm $P(i|\boldsymbol{x}) := \frac{e^{g(\boldsymbol{x})}}{1+e^{g(\boldsymbol{x})}} \in (0,1)$. As Table A1 shows, compared to $P(i|\boldsymbol{x})$, the regularized $P(i|\boldsymbol{x}) \cdot \hat{\beta}(\boldsymbol{x})$ even leads to worse OOD detection performance (*e.g.*, 33.6% *v.s.* 35.7% of FPR95). Hence, we propose to turn the probability-level regularization into logit-level, which is derived as $g^{\mathrm{bal}}(\boldsymbol{x}) := g(\boldsymbol{x}) + \log \hat{\beta}(\boldsymbol{x})$. The term $\log \hat{\beta}(\boldsymbol{x})$ plays the same role in calibrating the imbalanced $g(\boldsymbol{x})$ to the balanced $g^{\mathrm{bal}}(\boldsymbol{x})$, as discussed in Section 3.1. One may also take the sigmoid function to get the probability $P^{\mathrm{bal}}(i|\boldsymbol{x})$, whose numerical stability is validated in Table A1 as well. We thus develop the logit-level post-hoc normalization and training-time regularization techniques in Section 3.2 and Section 3.3, respectively.

Table A1: Post-hoc normalization for different OOD regularization on CIFAR10-LT.

| OOD Scorer | AUROC ($\uparrow$) | AUPR ($\uparrow$) | FPR95 ($\downarrow$) | ID ACC ($\uparrow$) |
|---|---|---|---|---|
| $P(i|\boldsymbol{x})$ | $87.37 \pm 0.23$ | $81.05 \pm 0.64$ | $33.66 \pm 0.12$ | $74.53 \pm 0.43$ |
| $P(i|\boldsymbol{x}) \cdot \hat{\beta}(\boldsymbol{x})$ | $87.18 \pm 0.31$ | $81.42 \pm 0.63$ | $35.78 \pm 0.71$ | $74.53 \pm 0.43$ |
| $g(\boldsymbol{x}) + \log \hat{\beta}(\boldsymbol{x})$ | $\mathbf{87.64} \pm 0.30$ | $\mathbf{81.61} \pm 0.63$ | $\mathbf{31.31} \pm 0.08$ | $74.53 \pm 0.43$ |

## B  ADDITIONAL EXPERIMENTAL RESULTS

### B.1  ADDITIONAL ERROR STATISTICS

In this section, we present the class-aware error statistics for OOD detection on different benchmarks (see Fig. A1) and different detectors (see Fig. A2). For each OOD detector on each benchmark, we first compute the OOD scores $g(\boldsymbol{x})$ for all the ID and OOD test data. Then, a threshold $\lambda$ is determined to ensure a high fraction of OOD data (*i.e.*, 95%) is correctly detected as out-of-distribution. Recall that $g(\boldsymbol{x})$ indicates the in-distribution probability for a given sample (*i.e.*, $P(i|\boldsymbol{x})$). Finally, we statistic the distributions of wrongly detected ID/OOD samples.

Specifically, in Fig. A1 and Fig. A2, the first row displays class labels of ID samples that are wrongly detected as OOD (*i.e.*, $g(\boldsymbol{x}) < \lambda$), and the second row exhibits class predictions of OOD samples that are wrongly detected as ID (*i.e.*, $g(\boldsymbol{x}) > \lambda$). In each subplot, we statistic the distribution over *head*, *middle*, and *tail* classes (the division rule follows Wang et al. (2022)) for simplicity. Note that the total count of wrong OOD samples (in the second row) is constant, and a better OOD detector $g$ will receive fewer wrong ID samples (in the first row).

Fig. A1 compares our method with PASCL (Wang et al., 2022) on CIFAR10/100-LT and ImageNet-LT benchmarks. The results indicate our method (dashed bar) performs relatively more balanced OOD detection on all benchmarks. We reduce the error number of ID samples from tail classes, and simultaneously decrease the error number of OOD samples that are predicted as head classes. In particular, our method achieves considerable balanced OOD detection on ImageNet-LT (the right column), which brings a significant performance improvement, as discussed in Section 4.2.

Fig. A2 compares our integrated version and vanilla OOD detectors (*i.e.*, OE, Energy, and BinDisc) on the CIFAR10-LT benchmark. Similarly, the results indicate our method performs relatively more balanced OOD detection against all original detectors. The versatility of our ImOOD framework is validated, as discussed in Section 4.3.

However, the statistics in Fig. A1 and Fig. A2 indicate the imbalanced problem has not been fully solved, since the scarce training samples of tail classes still affect the data-driven learning process. More data-level re-balancing techniques may be leveraged to further address this issue.

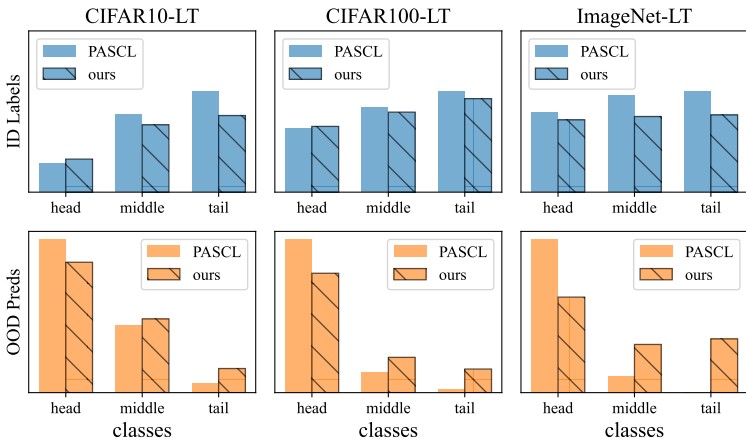

Figure A1: Class-aware *error* statistics for OOD detection on different benchmarks.

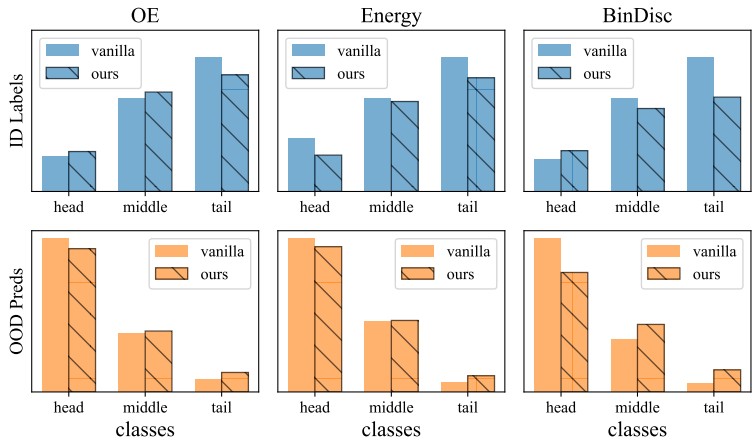

Figure A2: Class-aware *error* statistics for different OOD detectors on CIFAR10-LT.

## B.2 DETAILED RESULTS ON CIFAR10/100-LT BENCHMARKS

For CIFAR10/100-LT benchmark, Textures (Cimpoi et al., 2014), SVHN (Netzer et al., 2011), CIFAR100/10 (respectively), TinyImageNet (Le & Yang, 2015), LSUN (Yu et al., 2015), and Places365 (Zhou et al., 2017) from SC-OOD dataset Yang et al. (2021a) are adopted as OOD test sets. The mean results on the six OOD sets are reported in Section 4.

In this section, we reported the detailed measures on those OOD test sets in Table A2 and Table A3, as the supplementary to Table 2. The results indicate our method consistently outperforms the state-of-the-art PASCL (Wang et al., 2022) on most of the subsets.

Table A2: Detailed results on CIFAR10-LT.

| $\mathcal{D}_{out}^{test}$ | Method | AUROC ($\uparrow$) | AUPR ($\uparrow$) | FPR95 ($\downarrow$) |
|---|---|---|---|---|
| Texture | PASCL | $93.16 \pm 0.37$ | $84.80 \pm 1.50$ | $23.26 \pm 0.91$ |
| | **Ours** | $\mathbf{96.22} \pm 0.34$ | $\mathbf{93.50} \pm 0.79$ | $\mathbf{17.60} \pm 0.47$ |
| SVHN | PASCL | $96.63 \pm 0.90$ | $98.06 \pm 0.56$ | $12.18 \pm 3.33$ |
| | **Ours** | $\mathbf{97.12} \pm 0.90$ | $\mathbf{98.08} \pm 0.70$ | $\mathbf{10.47} \pm 3.58$ |
| CIFAR100 | PASCL | $84.43 \pm 0.23$ | $82.99 \pm 0.48$ | $57.27 \pm 0.88$ |
| | **Ours** | $\mathbf{86.03} \pm 0.09$ | $\mathbf{85.67} \pm 0.02$ | $\mathbf{50.84} \pm 0.34$ |
| TIN | PASCL | $87.14 \pm 0.18$ | $81.54 \pm 0.38$ | $47.69 \pm 0.59$ |
| | **Ours** | $\mathbf{89.07} \pm 0.35$ | $\mathbf{85.06} \pm 0.19$ | $\mathbf{40.34} \pm 1.62$ |
| LSUN | PASCL | $93.17 \pm 0.15$ | $91.76 \pm 0.53$ | $26.40 \pm 1.00$ |
| | **Ours** | $\mathbf{94.89} \pm 0.49$ | $\mathbf{94.38} \pm 0.45$ | $\mathbf{22.03} \pm 2.27$ |
| Places365 | PASCL | $91.43 \pm 0.17$ | $96.28 \pm 0.14$ | $33.40 \pm 0.88$ |
| | **Ours** | $\mathbf{93.06} \pm 0.31$ | $\mathbf{97.19} \pm 0.09$ | $\mathbf{28.30} \pm 1.76$ |
| **Average** | PASCL | $90.99 \pm 0.19$ | $89.24 \pm 0.34$ | $33.36 \pm 0.79$ |
| | **Ours** | $\mathbf{92.73} \pm 0.33$ | $\mathbf{92.31} \pm 0.12$ | $\mathbf{28.27} \pm 0.32$ |

Table A3: Detailed results on CIFAR100-LT.

| $\mathcal{D}_{out}^{test}$ | Method | AUROC ($\uparrow$) | AUPR ($\uparrow$) | FPR95 ($\downarrow$) |
|---|---|---|---|---|
| Texture | PASCL | $76.01 \pm 0.66$ | $58.12 \pm 1.06$ | $\mathbf{67.43} \pm 1.93$ |
| | **Ours** | $\mathbf{76.54} \pm 0.86$ | $\mathbf{62.38} \pm 1.22$ | $69.03 \pm 1.30$ |
| SVHN | PASCL | $80.19 \pm 2.19$ | $88.49 \pm 1.59$ | $53.45 \pm 3.60$ |
| | **Ours** | $\mathbf{84.77} \pm 0.39$ | $\mathbf{91.64} \pm 0.70$ | $\mathbf{46.56} \pm 1.29$ |
| CIFAR10 | PASCL | $\mathbf{62.33} \pm 0.38$ | $57.14 \pm 0.20$ | $\mathbf{79.55} \pm 0.84$ |
| | **Ours** | $62.09 \pm 0.59$ | $\mathbf{57.21} \pm 0.49$ | $80.92 \pm 0.48$ |
| TIN | PASCL | $\mathbf{68.20} \pm 0.37$ | $51.53 \pm 0.42$ | $76.11 \pm 0.80$ |
| | **Ours** | $68.08 \pm 0.50$ | $\mathbf{51.64} \pm 0.07$ | $\mathbf{75.92} \pm 0.84$ |
| LSUN | PASCL | $\mathbf{77.19} \pm 0.44$ | $\mathbf{61.27} \pm 0.72$ | $63.31 \pm 0.87$ |
| | **Ours** | $77.17 \pm 0.03$ | $61.08 \pm 0.92$ | $\mathbf{59.67} \pm 0.84$ |
| Places365 | PASCL | $76.02 \pm 0.21$ | $86.52 \pm 0.29$ | $64.81 \pm 0.27$ |
| | **Ours** | $\mathbf{76.14} \pm 0.29$ | $\mathbf{86.66} \pm 0.09$ | $\mathbf{62.31} \pm 0.46$ |
| **Average** | PASCL | $73.32 \pm 0.32$ | $67.18 \pm 0.10$ | $67.44 \pm 0.58$ |
| | **Ours** | $\mathbf{74.14} \pm 0.23$ | $\mathbf{68.43} \pm 0.53$ | $\mathbf{65.73} \pm 0.16$ |

## B.3 COMPARISON TO *BalEnergy*

BalEnergy (Choi et al., 2023) has recently achieved advance on CIFAR10/100LT benchmarks by adding an instance-specific loss regularization for each OOD training sample. However, its training and testing settings both differ from the PASCL (Wang et al., 2022) baseline. For example, PASCL trains all the models from scratch, while BalEnergy fine-tunes the OOD detector from a pre-trained classifier. During ID/OOD testing, PASCL treats ID samples as positive while BalEnergy treats OOD samples as positive, which causes a slight difference on the FPR95 measure (Tao et al., 2023b).

Thus, we separately evaluate our method on BalEnergy's settings for a fair comparison. The results in Table A4 demonstrate that our ImOOD can still effectively boost the OOD detection performance.

As we discussed in Section 4.3, the combination with ImOOD and advanced techniques may bring better OOD detectors.

Table A4: Comparison with *BalEnergy* (Choi et al., 2023).

| Method | AUROC (↑) | AUPR (↑) | FPR95 (↓) | ID ACC (↑) |
|---|---|---|---|---|
| ID Dataset: CIFAR10-LT | | | | |
| BalEnergy | $92.51 \pm 0.13$ | $91.87 \pm 0.11$ | $31.11 \pm 0.24$ | $76.00 \pm 0.21$ |
| **ImOOD** (Ours) | $\mathbf{92.80} \pm 0.15$ | $\mathbf{92.26} \pm 0.18$ | $\mathbf{30.29} \pm 0.19$ | $\mathbf{76.47} \pm 0.31$ |
| ID Dataset: CIFAR100-LT | | | | |
| BalEnergy | $77.55 \pm 0.08$ | $72.95 \pm 0.12$ | $\mathbf{61.54} \pm 0.25$ | $40.94 \pm 0.11$ |
| **ImOOD** (Ours) | $\mathbf{77.85} \pm 0.12$ | $\mathbf{73.96} \pm 0.11$ | $61.82 \pm 0.18$ | $\mathbf{40.99} \pm 0.08$ |

