# OpenReview forum: "Rethinking Out-of-Distribution Detection on Imbalanced Data Distribution"
_ICLR.cc/2024/Conference — Submitted to ICLR 2024_

### Official Review · Reviewer_b8DW · 2023-10-22

**Soundness:** 3 good
**Presentation:** 3 good
**Contribution:** 2 fair
**Rating:** 6
**Confidence:** 4

**Summary:**

The paper studies the problem out-of-distribution (OOD) detection when original in-distribution (ID) data suffers from class imbalance. The authors first report an experimental observation regarding two major challenges in OOD detection under the studied setting: misidentification of tail-side ID samples as OOD (False positives) and of some OOD samples as head-side ID (False negatives). They then statistically compare the learning objectives in OOD detection with class-balanced vs. -imbalanced ID data, allowing them to attribute these challenges to the auxiliary bias term that does not cancel out in the class-imbalanced scenario. Based on this analysis, they propose two unique avenues to mitigate the influence of this bias term: post-hoc normalization and train-time regularization. Together, they form a cohesive framework for OOD detection with class-imbalanced ID data, dubbed ImOOD. The authors provide substantial empirical evidence for the effectiveness of ImOOD.

**Strengths:**

- The statistical analysis performed in the paper is sound and written in a manner that is easy to follow. The authors include a helpful intuitive understanding of the bias term that connects it back to the empirical observations about the behavior of OOD detection models under the class-imbalanced setting.
- ImOOD is well-motivated and theoretically-grounded on the the above analysis. The general thinking behind the method is sound, and for the most part, it’s described clearly enough to understand and re-implement the design.

**Weaknesses:**

- I believe the major drawback of this paper lies in the limited scope of the studied problem. It appears that the authors assume that the class-imbalance problem only exists on the original ID dataset. I am not too convinced with the practicality of the proposed scenario; it’s hard to imagine in what deployment scenarios only the ID dataset will suffer from this problem, while the OOD dataset(s) used for training/testing is void of it. Wouldn’t it be more natural to assume that the ID training dataset can be refined in advance to make it class-balanced, while the OOD data the detector must be able to identify arise in various forms, and thus is more likely to be class-imbalanced?
    - What if the auxiliary OOD dataset used for training and/or the target OOD dataset exhibits class-imbalance as well? I think it is important to study various combinations of class-balanced and -imbalanced ID/auxiliary OOD/target OOD datasets. Does the analysis performed in the paper extend to and hold in such settings? Can ImOOD still outperform other baselines?
- Please correct me if I am wrong, but it appears that the verification of post-hoc normalization (minus the train-time regularization) on larger datasets appears to be missing. Even on CIFAR10-LT, the improvement from post-hoc normalization seems marginal, but this concern could be alleviated, as long as the improvement is consistent across various datasets. If post-hoc normalization is not quite as effective on datasets, it would signify that ImOOD is heavily reliant on the train-time regularization.
- More details on how the label distribution is learned (during test-time regularization) would be appreciated. Also, do you use the same learned label distribution when performing post-hoc normalization? Or is it used only for training-time regularization? If you discard it after training, one would have to know the label distribution anyways for post-hoc normalization, so do we really need to use the learned distribution in the first place?
- The empirical validation is limited to one target OOD dataset per InD dataset. Validation on more challenging OOD datasets (e.g., near OOD data, data with spurious correlation) would be helpful to gauge the effectiveness of the proposed method.

**Questions:**

Please refer to the Weaknesses section.

---

> ### Author Response · Authors · 2023-11-19
> **Response to reviewer b8DW (Part. I)**
>
> We thank the valuable comments, and we answer them in detail as follows.
>
> > Q1: How to deal with the self-imbalanced OOD data?
>
> The self-imbalanced OOD data is indeed a practical problem.
> As suggested, we now take CIFAR100 as the imbalanced auxiliary OOD data, and train models with CIFAR10 as ID.
> Then models are evaluated on the remaining 5 subsets of the SCOOD benchmark (CIFAR100 was originally used as test OOD data but is now eliminated).
> According to the results reported below, our ImOOD consistently outperforms the advanced PASCL method.
>
> |  Method  |   AUROC↑  |    AUPR↑  |   FPR95↓  |
> |:--------:|:---------:|:---------:|:---------:|
> |   PASCL  |   88.05   |   86.63   |   38.69   |
> | **Ours** | **89.14** | **87.30** | **35.48** |
>
> Besides, we have also noticed a recent work by BalEnergy $^{[1]}$ that discussed this issue and added class-aware loss regularizations on OOD training data.
> According to the experiments in Appendix B.3 (due to different experimental settings) of our submitted paper, we equipped our ImOOD with BalEnergy and achieved better OOD detection performance.
> It indicates our method does not conflict with imbalanced OOD data optimization techniques.
>
> |   ID Data   |   Method  |   AUROC↑  |   AUPR↑   |   FPR95↓  |
> |:-----------:|:---------:|:---------:|:---------:|:---------:|
> |  CIFAR10-LT | BalEnergy |   92.51   |   91.87   |   31.11   |
> |             | **+Ours** | **92.80** | **92.26** | **30.29** |
> | CIFAR100-LT | BalEnergy |   77.55   |   72.95   |   61.54   |
> |             | **+Ours** | **77.85** | **73.96** |   61.82   |
>
> > Q2: About the effectiveness of post-hoc normalization.
>
> Thanks for pointing out this. In the manuscript, we only provide the performance gain brought by our post-hoc normalization on CIFAR10-LT benchmark. Here we supplement the results on CIFAR100-LT and ImageNet-LT benchmarks, which shows applying our method leads to consistent improvements on various datasets. The general validation of our post-hoc normalization is verified.
>
> |  Benchmark  |   Method  |   AUROC↑  |   AUPR↑   |   FPR95↓  |
> |:-----------:|:---------:|:---------:|:---------:|:---------:|
> |  CIFAR10-LT |  BinDisc  |   87.37   |   81.05   |   33.66   |
> |             | **+Ours** | **87.64** | **81.61** | **31.31** |
> | CIFAR100-LT |  BinDisc  |   72.09   |   67.63   |   68.70   |
> |             | **+Ours** | **72.33** | **67.96** | **67.54** |
> | ImageNet-LT |  BinDisc  |   70.12   |   69.64   |   78.13   |
> |             | **+Ours** | **70.74** | **70.28** | **76.83** |
>
> However, as our experiments suggest, the enhancement by post-hoc normalization is relatively limited, while our training-time regularization seems to bring more significant advances.
> The main reason may be that the estimates of probability distributions are not well-calibrated, especially for the class prior $P(y) \coloneqq \pi_y$ whose numerical instability may severely affect the OOD detection process, as discussed in Appendix 2.3 in the submitted paper.
> We thus further develop the training-time regularization technique to automatically adjust the estimates, in order to ultimately learn a better OOD detector $g(x)$ close to the ideally balanced $g^{bal}(x)$.
> Once $g(x)$ is optimized, no estimate is used for detecting OOD samples, and the output of scorer $g(x)$ is more stable and effective in performing OOD detection.
>
> We will add the experiments and discussions.

---

> > ### Author Response · Authors · 2023-11-19
> > **Response to reviewer b8DW (Part. II)**
> >
> > > Q3: Details about the label prior $P(y)$.
> >
> > We are sorry to make you confused. During **training**, we make the initial statistic estimates $P(y) \coloneqq \pi_y$ as **learnable** $\pi_y(\theta)$, **merely** to better push $g(x)$ close to the ideal $g^{bal}(x)$.
> > During **testing**, the estimated $\pi_y(\theta)$ will be **discarded**, and we only adopt the already optimized $g(x)$ to perform OOD detection.
> > In this way, people do not need to know the learned $\pi_y(\theta)$, as no more post-hoc normalization is required.
> >
> > We will better clarify the whole procedure of our method.
> >
> > > Q4: Evaluation on different OOD test sets.
> >
> > In the paper we mainly compare different OOD detectors on the SCOOD benchmark $^{[2]}$, a composite evaluation dataset with 6 subsets covering different scenarios.
> > Here we identify two representative subsets from SCOOD to further demonstrate our efficacy.
> > Specifically, SVHN can be viewed as far OOD, and CIFAR100 can be seen as near OOD (with CIFAR10-LT as ID), as suggested by Fort et al $^{[3]}$.
> > As we previously discussed, our ImOOD brings consistent enhancement against the strong baseline PASCL, especially on CIFAR100 (near OOD) test set with a decrease of 6.4% on FPR95.
> >
> > In addition, we also report the spurious OOD detection evaluation as suggested by Ming et al $^{[4]}$. The WaterBird ID dataset also suffers from the imbalance problem (on water birds and land birds), and Ming et al specifically collected a subset of Places as the spurious OOD test set (with spurious correlation to background). The results below also demonstrate our method's robustness in dealing with spurious OOD problems.
> >
> > |     Evaluation Dataset    |  Method  |   AUROC↑  |    AUPR↑  |   FPR95↓  |
> > |:-------------------------:|:--------:|:---------:|:---------:|:---------:|
> > |      SCOOD Benchmark      |   PASCL  |   90.99   |   89.24   |   33.36   |
> > |                           | **Ours** | **92.73** | **92.31** | **28.27** |
> > |       SVHN (Far OOD)      |   PASCL  |   96.63   |   98.06   |   12.18   |
> > |                           | **Ours** | **97.12** | **98.08** | **10.47** |
> > |    CIFAR100 (Near OOD)    |   PASCL  |   84.43   |   82.99   |   57.27   |
> > |                           | **Ours** | **86.03** | **85.67** | **50.84** |
> > | WaterBird (Spurious OOD)  |   PASCL  |   89.59   |   91.05   |   33.73   |
> > |                           | **Ours** | **90.63** | **92.49** | **30.11** |
> >
> >
> > [1] Choi et al. Balanced energy regularization loss for out- of-distribution detection. CVPR, 2023.
> >
> > [2] Yang et al. Semantically Coherent Out-of-Distribution Detection. ICCV, 2021.
> >
> > [3] Fort et al. Exploring the Limits of Out-of-Distribution Detection. NeurIPS, 2021.
> >
> > [4] Ming et al. On the Impact of Spurious Correlation for Out-of-distribution Detection. AAAI, 2022.

---

### Official Review · Reviewer_LaLW · 2023-10-28

**Soundness:** 3 good
**Presentation:** 3 good
**Contribution:** 2 fair
**Rating:** 6
**Confidence:** 4

**Summary:**

When encountering inherent imbalance of in-distribution (ID) data, the paper identified two common challenges faced by different OOD detectors: misidentifying tail class ID samples as OOD, while erroneously predicting OOD samples as head class from ID. To explain this phenomenon, the authors introduce a generalized statistical framework, termed ImOOD, to formulate the OOD detection problem on imbalanced data distribution. Consequently, the theoretical analysis reveals that there exists a class-aware bias item between balanced and imbalanced OOD detection, which contributes to the performance gap.

**Strengths:**

1. The paper is written well and is easy to understand.
2. The studied problem is very important.
3. The results seem to outperform state-of-the-art.

**Weaknesses:**

1. I am curious about the possibility of this method being extended to distance-based OOD detection methods, such as Mahalanobios distance, etc. How well does the current method compare to the state-of-the-art distance-based methods?
2. I am curious about how well the current method performs on synthesized outliers (Tao et.al. 2023).
3. Could you elaborate more on a closely related baseline, Jiang et al. 2023, and discuss the similarities and differences w.r.t your work?

**Questions:**

see above

---

> ### Author Response · Authors · 2023-11-19
> **Response to reviewer LaLW (Part. I)**
>
> We thank the valuable comments, and we answer them in detail as follows.
>
> > Q1: Comparison and extension to distance-based OOD methods.
>
> Thanks for the thoughtful suggestion. Due to the time limitation, here we mainly compare our method with Mahalanobios distance $^{[1]}$ (denoted as "Maha") on the CIFAR10-LT benchmark. The results show that our post-hoc normalization (denoted as "Maha + infer") brings slight improvements (e.g., 0.3% increase on AUROC), while our training-time regularization (denoted as "Maha + train") significantly boosts the imbalanced OOD detection (e.g., 1.2% increase on AUROC and 3.4% decrease on FPR95).
>
> |    Method    |   AUROC↑  |   AUPR↑   |   FPR95↓  |
> |:------------:|:---------:|:---------:|:---------:|
> |     Maha     |   88.26   |   87.94   |   42.74   |
> | Maha + infer |   88.58   |   87.98   |   42.47   |
> | Maha + train | **89.43** | **88.31** | **39.37** |
>
> However, the final results are still worse than advanced models (with 92%+ on AUROC for example). A possible reason might be that the feature-level distance estimation is basically inaccurate, especially for the tail classes with rare data samples to statistic (e.g., computing the mean and variance). On the other hand, our method takes the imbalance problem into consideration, which helps balance the feature space by using the training-time regularization.
>
> > Q2: Ablation on synthesized OOD training data.
>
> Thanks for the insightful suggestion. Here we equip with the novel VOS $^{[2]}$ and NPOS $^{[3]}$ to synthesize virtual OOD data in the feature space for training.
> According to the results shown below, synthesized outliers can also improve the OOD detection performance (e.g., 72% v.s. 75% on AUROC), and our method brings further advance regarding the vanilla VOS and NPOS methods (e.g., 75% v.s. 78% on AUROC).
>
>
> | Training OOD |  Method  |   AUROC↑  |    AUPR↑  |   FPR95↓  |
> |:------------:|:--------:|:---------:|:---------:|:---------:|
> |     None     |    MSP   |   72.09   |   70.40   |   68.79   |
> |    VOS-Syn   |    VOS   |   75.44   |   73.86   |   65.04   |
> |              | **Ours** | **78.61** | **76.97** | **64.65** |
> |   NPOS-Syn   |   NPOS   |   74.83   |   74.02   |   69.67   |
> |              | **Ours** | **77.05** | **76.52** |   70.85   |
>
> However, compared to the real OOD training data (e.g., the TinyImages80M used in our original setting), the synthesized outliers seem to be unsatisfactory to help train a SOTA OOD detector. It indicates that the density-based VOS and the distance-based NPOS also suffer from inaccurate feature-level statistics on imbalanced data distribution, and are unable to generate qualified virtual outliers.
> This is more severe for models training from scratch, as NPOS also discussed, and image-level OOD sample synthesis may help alleviate this problem.
>
> We will add those experiments and discussions.

---

> > ### Author Response · Authors · 2023-11-19
> > **Response to reviewer LaLW (Part. II)**
> >
> > > Q3: Detailed discussion with the recent related work ClassPrior $^{[4]}$.
> >
> > Thanks for pointing out this. Here we summarize the main similarity and differences regarding ClassPrior in detail.
> >
> > The major **similarity**, as discussed in the submitted manuscript, is that both ClassPrior and our method incorporate ID class prior to boosting imbalanced OOD detection. On the other hand, the **differences** mainly lie in three aspects:
> >
> > 1. The **motivation** between our ImOOD and ClassPrior is different. ClassPrior assumes a different posterior distribution $P(y|x)$ on ID data and OOD data, and manually develops two ad-hoc distance-measures regarding $P(y|x)$ and $P(y)$ to adjust existing OOD scorers without **theoretical support**. Instead, our method theoretically analyzes the bias between imbalanced OOD detector $g(x)$ and ideal (balanced) $g^{bal}(x)$, based on which we further provide a unified view of post-hoc normalization and training-time regularization techniques to push $g(x)$ towards $g^{bal}(x)$.
> > 2. ClassPrior makes a **strong assumption** that the posterior probability of OOD data equals the prior ID probability, that is, $P(y|x \in \mathcal{D}^{out}) = P(y)$ (see its Theorem 3.2). The question is this assumption does not always hold. In our Table 2, we report its "RP+MSP" (RePlacing) strategy on models trained with CIFAR10-LT (with ID data only), which leads to negligible improvements (e.g., 0.3% increase of AUROC). We further test its "RW+Energy" (ReWeighting) strategy on the same model, as shown in the table below, which even causes performance degradation. In contrast, our method does not make any assumption on $P(y|x \in \mathcal{D}^{out})$, and shows strong robustness to the practical scenarios (e.g., 4.3% increase of AUROC).
> >
> > |  Post-hoc  | AUROC↑ |  AUPR↑ | FPR95↓ |
> > |:----------:|:-----:|:-----:|:-----:|
> > |    Energy  | 73.16 | 71.73 | 69.19 |
> > | ClassPrior | 72.22 | 70.88 | 70.75 |
> > |    **Ours**    | **77.48** | **76.29** | **66.47** |
> >
> > 3. Our ImOOD provides **training-time regularization** to further boost the imbalanced OOD detection, where ClassPrior is **inapplicable** since its assumption $P(y|x \in \mathcal{D}^{out}) = P(y)$ does not necessarily hold during training (e.g., OE assumes $P(y|x \in \mathcal{D}^{out}) = \frac{1}{N_c} \neq P(y)$).
> >
> > We will supplement those discussions to clarify our novelty and contribution.
> >
> > [1] Lee et al. A Simple Unified Framework for Detecting Out-of-Distribution Samples and Adversarial Attacks. NeurIPS, 2018.
> >
> > [2] Du et al. Learning What You Don’t Know by Virtual Outlier Synthesis. ICLR, 2022.
> >
> > [3] Tao et al. Non-Parametric Outlier Synthesis. ICLR, 2023.
> >
> > [4] Jiang et al. Detecting Out-of-distribution Data through In-distribution Class Prior. ICML, 2023.

---

### Official Review · Reviewer_r7k6 · 2023-10-31

**Soundness:** 3 good
**Presentation:** 3 good
**Contribution:** 3 good
**Rating:** 6
**Confidence:** 4

**Summary:**

To tackle the OOD detection under class imbalanced setting, the paper proposed the statistically guided framework called ImOOD. Through the statistical analysis, the authors have found that there exists a bias term responsible for the performance gap between balanced and imbalanced OOD detection models. Under the ImOOD framework, by leveraging the bias term, the authors then propose the post-hoc normalization and training time regularization technique to enhance the OOD performance. The experimentation conducted on multiple real-world datasets demonstrates the effectiveness of the proposed post-hoc normalization and training time regularization techniques.

**Strengths:**

* The motivation for proposing the ImOOD framework is clear. The authors have done a great job in terms of empirically demonstrating the limitations present in existing OOD detection techniques under imbalanced data distribution. For example,  Figure 1 demonstrates how OOD samples are incorrectly detected as head class samples and that of the in-distribution tail class samples as OOD samples.
* The proposed post-hoc normalization and training time regularization are backed by the strong statistical analysis conducted in Section 3.1 along with empirical evidence.
* The paper is well-written and easy to follow.
* An extensive ablation study is conducted to showcase the effectiveness of each component in their proposed framework. For example, in Table 4 the authors have shown how the estimation of the class-prior in the training-time regularization helps to improve the performance.

**Weaknesses:**

* The authors have used the auxiliary OOD training dataset and its effect on performance is not very clear. It would be interesting to see the sensitivity of the OOD performance with respect to the selection of different OOD training data. For example, what does the performance look like if we use the Cifar100 as OOD data for the model with Cifar10 as ID data?
* The authors may need to report the performance of the wide range of OOD detection methods to demonstrate the effectiveness of their proposed post-hoc normalization and training time regularization. For example, the authors may consider the most representative OOD techniques like OpenMAX [1], CGDL [2], OLTR [3], etc.
* The performance gain using the post hoc normalization seems to be marginal on OE and BinDisc, especially in terms of AUROC and AUPR. Having a more detailed explanation for this would be useful.

**References**
1. Bendale et al. “Towards open set deep networks”. CVPR2016.
2. Sun et al. “Conditional gaussian distribution learning for open set recognition”. CVPR2020
3. Liu et al. “Large-scale long-tailed recognition in an open world.” CVPR2019.

**Questions:**

It would be interesting to see the experimental results mentioned in the weaknesses section.

---

> ### Author Response · Authors · 2023-11-19
> **Response to reviewer r7k6 (Part. I)**
>
> We thank the valuable comments, and we answer them in detail as follows.
>
> > Q1: Ablation on the choice of auxiliary OOD training data.
>
> Thanks for the insightful suggestion. In the main experiments reported in our manuscript, we follow PASCL $^{[1]}$ to adopt the TinyImages80M $^{[2]}$ as auxiliary OOD data for training-time regularization, and the models are tested on the SCOOD benchmark for 6 subsets.
> We now train the models with CIFAR100 as auxiliary OOD data (CIFAR10-LT is ID), and test them on the remaining 5 subsets of the SCOOD benchmark $^{[3]}$ (CIFAR100 was originally used as test OOD data but is now eliminated).
> According to the results reported below, our ImOOD consistently outperforms the advanced PASCL method.
>
> |  Method  |   AUROC↑  |    AUPR↑  |   FPR95↓  |
> |:--------:|:---------:|:---------:|:---------:|
> |   PASCL  |   88.05   |   86.63   |   38.69   |
> | **Ours** | **89.14** | **87.30** | **35.48** |
>
> However, models trained with CIFAR100 perform slightly worse than models trained with TinyImages80M, indicating that a more generally collected auxiliary OOD dataset also contributes to the final OOD detection performance, which is consistent with common sense.
> We will add the experiment and discussion.
>
> > Q2: Comparison with other representative OSR/OOD detection methods.
>
> Thanks for the thoughtful suggestion. Due to the time limitation, we mainly compare our method with the representative OpenMAX $^{[5]}$ and OLTR $^{[6]}$ on CIFAR100-LT benchmark.
>
> Before the comparison, we want to carefully identify that the classical Open-Set Recognition (OSR) setting is slightly different from the Out-Of-Distribution Detection (OOD) setting $^{[4]}$. In particular, vanilla OSR methods (like OpenMAX and OLTR) take a proportion (e.g., 50%) of CIFAR100-LT classes as ID (or, base) classes, and the remaining as OOD (or, novel) classes. No auxiliary OOD training data is utilized for optimization. To ensure comparability, we follow the OOD setting, that is, taking all 100 classes from CIFAR100-LT as ID classes, and use the corresponding [codebase](https://github.com/ma-xu/Open-Set-Recognition) to train the models of OpenMAX and OLTR without auxiliary OOD dataset. OOD test set is the SCOOD benchmark.
>
> The results below show that compared to the MSP baseline, OpenMAX gains 1.4% on AUROC, and OLTR further improves 1.0% on AUROC and 3.5% on AUPR by taking the imbalanced distribution into account. Moreover, with the same model as MSP baseline, Energy shows comparable performance, and our method on post-hoc normalization further brings significant improvement, reaching 66.38% of AUROC and 76.98% of FPR95. Finally, we also report the performance of our method trained with auxiliary OOD data (i.e., TinyImages80M) merely as a reference, to avoid unfair comparison with the above methods without OOD training data.
>
>
> |  Method  |  Aux OOD Data  |    AUROC↑   |    AUPR↑    |    FPR95↓   |
> |:--------:|:--------------:|:-----------:|:-----------:|:-----------:|
> |    MSP   |       No       |    62.17    |    57.99    |    84.14    |
> |  OpenMAX |       No       |    63.58    |    57.91    |    80.40    |
> |   OLTR   |       No       |    64.52    |    60.43    |    82.88    |
> |  Energy  |       No       |    64.87    |    61.11    |    78.89    |
> | **Ours** |       No       |  **66.38**  |  **61.76**  |  **76.98**  |
> | **Ours** |    **_Yes_**   | **_74.14_** | **_68.43_** | **_65.73_** |

---

> > ### Author Response · Authors · 2023-11-19
> > **Response to reviewer r7k6 (Part. II)**
> >
> > > Q3: About the effectiveness of post-hoc normalization.
> >
> > Thanks for pointing out this. In the manuscript, we only provide the performance gain brought by our post-hoc normalization on CIFAR10-LT benchmark. Here we supplement the results on CIFAR100-LT and ImageNet-LT benchmarks, which shows applying our method leads to consistent improvements on various datasets. The general validation of our post-hoc normalization is verified.
> >
> > |  Benchmark  |   Method  |   AUROC↑  |   AUPR↑   |   FPR95↓  |
> > |:-----------:|:---------:|:---------:|:---------:|:---------:|
> > |  CIFAR10-LT |  BinDisc  |   87.37   |   81.05   |   33.66   |
> > |             | **+Ours** | **87.64** | **81.61** | **31.31** |
> > | CIFAR100-LT |  BinDisc  |   72.09   |   67.63   |   68.70   |
> > |             | **+Ours** | **72.33** | **67.96** | **67.54** |
> > | ImageNet-LT |  BinDisc  |   70.12   |   69.64   |   78.13   |
> > |             | **+Ours** | **70.74** | **70.28** | **76.83** |
> >
> > However, as our experiments suggest, the enhancement by post-hoc normalization is relatively limited, while our training-time regularization seems to bring more significant advances.
> > The main reason may be that the estimates of probability distributions are not well-calibrated, especially for the class prior $P(y) \coloneqq \pi_y$ whose numerical instability may severely affect the OOD detection process, as discussed in Appendix 2.3 in the submitted paper.
> > We thus further develop the training-time regularization technique to automatically adjust the estimates, in order to ultimately learn a better OOD detector $g(x)$ close to the ideally balanced $g^{bal}(x)$.
> > Once $g(x)$ is optimized, no estimate is used for detecting OOD samples, and the output of scorer $g(x)$ is more stable and effective in performing OOD detection.
> >
> > We will add the experiments and discussions.
> >
> > [1] Wang et al. Partial and Asymmetric Contrastive Learning for Out-of-Distribution Detection in Long-Tailed Recognition. ICML, 2022.
> >
> > [2] Torralba et al. 80 million tiny images: A large data set for nonparametric object and scene recognition. TPAMI, 2008.
> >
> > [3] Yang et al. Semantically Coherent Out-of-Distribution Detection. ICCV, 2021.
> >
> > [4] Cen et al. The Devil is in the Wrongly-classified Samples: Towards Unified Open-set Recognition. ICLR, 2023.
> >
> > [5] Bendale et al. Towards open set deep networks. CVPR, 2016.
> >
> > [6] Liu et al. Large-scale long-tailed recognition in an open world. CVPR, 2019.

---

> > ### Comment · Reviewer_r7k6 · 2023-11-21
> > **Comment on author response**
> >
> > I appreciate the thorough rebuttal. The authors have addressed most of my concerns and would like to keep my current score. However, I still feel that it is crucial to carefully choose auxiliary OOD data during the training, taking into account the specific nature of the OOD data that may arise in the testing phase.  As such, the applicability of the proposed technique in the real-world OOD detection task can be rather limited.

---

> > > ### Author Response · Authors · 2023-11-22
> > > **Thanks for reviewer r7k6's feedback**
> > >
> > > We sincerely thank the reviewer again for evaluating our work. We will stick to real-world applications in our future work.
> > >
> > > Best regards,
> > >
> > > Authors

---

### Official Review · Reviewer_qdXa · 2023-11-01

**Soundness:** 2 fair
**Presentation:** 3 good
**Contribution:** 2 fair
**Rating:** 5
**Confidence:** 3

**Summary:**

The main contribution of this paper is the introduction of a generalized statistical framework called ImOOD, which addresses the problem of detecting and rejecting unknown out-of-distribution (OOD) samples in the presence of imbalanced data distributions. The paper identifies two common challenges faced by existing OOD detection methods: misclassifying tail class in-distribution (ID) samples as OOD, and incorrectly predicting OOD samples as head class ID samples. Then, the authors propose a general framework that can lead to improved detection performance.

**Strengths:**

The paper introduces a generalised statistical framework called ImOOD, which addresses the problem of OOD detection in the presence of imbalanced data distributions. Imbalanced data distributions widely exist in real world, and a set of fundamental works have proposed to tackle this questions to boost OOD detection. It seems that there may still exist many open questions, making the research direction in this paper promising in OOD detection.

The paper identifies two common challenges faced by existing OOD detection methods, namely misclassifying tail class ID samples as OOD and incorrectly predicting OOD samples as head class ID samples. This identification helps in understanding the limitations of current approaches.

The proposed method  is driven by the Bayesian analysis, demonstrating the impacts of data imbalance and suggesting a general framework that can handle the distribution shift in OOD detection. The proposed framework, as claimed by the authors, can be used to handle imbalanced issue for a set of different scoring strategies, and the evaluation results further verify the power of their method against imbalance data more or less.

**Weaknesses:**

A direct question is that if we have used learning algorithms that can handle imbalanced data (which is often the case in reality) to train the basic model, do we need to handle imbalanced data for OOD detection thereafter.

The authors identify two cases that make previous OOD detection methods fail, i.e., misclassifying tail class ID samples as OOD and incorrectly predicting OOD samples as head class ID samples. It is a direct conclusion and seemingly to be important, but could the authors further connect these two cases to previous works that handle imbalanced data in OOD detection and the proposed ImOOD. For example, why previous works, such as [1], will fail to discern these two cases and why ImOOD can overcome previous drawbacks.

g should be defined in a proper position in advance. It seems that g should be the detector built upon f.

My main concern lies in the inaccurate estimation of beta, consisting of three terms that are all biased from my view. For P(i|X), a direct failure case is that when we have a detector whose g always greater than 0 (e.g., for distance based scoring), then P(i|X) will be always greater than 0.5. Thus, all data points will be taken as ID cases, making the estimation obviously a biased case. P(y|x,i) also suffers from biased estimation, due to the well known calibration failure of deep models. Such an issue, from my view, cannot be ignored in the field of OOD detection, since it is one of the main reason why MSP score is not effective in practice. The estimation of P(y) is also biased, since the training time data distribution (ID + OOD) is different from the test situation, thus n_o used for training cannot cover the real test situation.


Detailed discussion about the hyper-parameters setting and the choice of auxiliary OOD data and the evaluation datasets should be discussed in detailed. More methods that handle OOD detection with imbalanced data should be considered here, such as [1]. More ablation studies to test the respective power of P(i|x), P(y), and P(y|x) should be tested.


[1] Detecting Out-of-distribution Data through In-distribution Class Prior.

**Questions:**

Please see the Weaknesses.

---

> ### Author Response · Authors · 2023-11-19
> **Response to reviewer qdXa (Part. I)**
>
> We thank the valuable comments, and we answer them in detail as follows.
>
> > Q1: Can algorithms that merely handle the imbalanced ID classification immediately address the OOD detection issue?
>
> Thanks for pointing out the important preliminary problem. Wang et al.$^{[1]}$ had widely discussed this issue and conducted a series of experiments (part of the results on CIFAR10-LT is displayed below). The results indicated that simply combining the ID classification optimization (e.g., LA $^{[2]}$ or $\tau$-norm $^{[3]}$) with vanilla OOD detection techniques (e.g., OE) cannot address the challenge. We thus have to take imbalanced ID classification and OOD detection into integrated consideration.
>
> |   Method  | ARUOC↑ |  AUPR↑ | FPR95↓ |
> |:---------:|:-----:|:-----:|:-----:|
> |  OE + **None**  | _89.92_ | _87.71_ | 34.80 |
> | OE + $\tau$-norm | 89.58 | 85.88 | _33.80_ |
> |   OE + LA   | 89.46 | 86.39 | 34.94 |
> |    **Ours**   | **92.73** | **92.31** | **28.27** |
>
> Relevant citations and conclusions will be added to the revised manuscript.
>
> > Q2: Why previous methods failed at handling the imbalance problem but our ImOOD works?
>
> Thanks for the thoughtful suggestion. We briefly summarize the main drawbacks of prious representative methods (PASCL $^{[1]}$ and ClassPrior $^{[4]}$):
>
> 1. **Our method** concurrently considers the misidentified ID samples from tail-class and OOD data predicted as head-class, as discussed in Section 3.1 in our manuscript.
> 2. **PASCL** focuses on improving the separability between tail-class ID samples and OOD data, while **ignores** the misidentification between OOD data and ID head-class, as illustrated in Figure 1 in our manuscript.
> 3. **ClassPrior** takes both the two factors into account, and develops a series of manually-designed post-hoc normalization techniques on models only trained with ID data. However, it makes a **strong assumption** that the posterior probability of OOD data equals the prior ID probability, that is, $P(y|x \in \mathcal{D}^{out}) = P(y)$ (see its Theorem 3.2). The question is this assumption does not always hold. In our Table 2, we report its "RP+MSP" (RePlacing) strategy on models trained with CIFAR10-LT, which leads to negligible improvements (e.g., 0.3% increase of AUROC). We further test its "RW+Energy" (ReWeighting) strategy on the same model, as shown in the table below, which even causes performance degradation. **In contrast**, our method does not make any assumption on $P(y|x \in \mathcal{D}^{out})$, and shows strong robustness to the practical scenarios (e.g., 4.3% increase of AUROC). **Besides**, our ImOOD provides training-time regularization to further boost the imbalanced OOD detection, where ClassPrior is inapplicable since its assumption $P(y|x \in \mathcal{D}^{out}) = P(y)$ does not necessarily hold during training.
>
> |  Post-hoc  | AUROC↑ |  AUPR↑ | FPR95↓ |
> |:----------:|:-----:|:-----:|:-----:|
> |    Energy  | 73.16 | 71.73 | 69.19 |
> | ClassPrior | 72.22 | 70.88 | 70.75 |
> |    **Ours**    | **77.48** | **76.29** | **66.47** |
>
> We will supplement those discussions to clarify our novelty and contribution.

---

> > ### Author Response · Authors · 2023-11-19
> > **Response to reviewer qdXa (Part. II)**
> >
> > > Q3: About the estimation of $P(i|x)$, $P(y|x,i)$, and $P(y)$.
> >
> > Thanks for the insightful comment. We want to clarify that our ImOOD's main contribution is the theoretically grounded framework to push the imbalanced OOD detector $g(x)$ close to the ideal (balanced) detector $g^{bal}(x)$. Our manuscript provides practical estimates of $P(i|x)$, $P(y|x,i)$, and $P(y)$ as an empirically effective approach to achieving this goal.
> > Below we discuss the estimation of the three items respectively:
> >
> > 1. **About $g(x)$ and $P(i|x)$.** As stated in Appendix A.2, $g(x)$ is implemented with an **extra linear layer** upon various OOD scores like Energy, as $g(x) = w \cdot S(x;f) + b$. Thus, for a distance-based scorer $S$, though $S(x;f) > 0$ may hold, the value of $g(x)$ is NOT always greater than 0 (thanks to item $b$ specifically), which means the estimated $P(i|x) \in (0,1)$ is also NOT biased.
> > We will add a concise statement in the main body of our manuscript as you suggested.
> >
> > 2. **About $P(y|x,i)$.** Using the softmax function to project NN's logit outputs into probability distribution is widely used in the imbalanced recognition and OOD detection literature, such as Menon et al. $^{[2]}$ and Jiang et al $^{[4]}$.
> > We acknowledge there exists a gap between the estimated distribution and the ground-truth (generally unknown) distribution, which is exactly the reason why we unify the softmax outputs into the whole optimization process, rather than adopting the maximum softmax probability (MSP) as the direct scorer.
> > We view the strict model calibration as our future work.
> >
> > 3. **About $P(y)$.** During training, we make the initial statistic estimates $P(y) \coloneqq \pi_y$ as adjustable $\pi_y(\theta)$, **merely** to better push $g(x)$ closer to the ideal $g^{bal}(x)$.
> > During testing, the estimated $\pi_y(\theta)$ will be **discarded**, and we only adopt the already optimized $g(x)$ to perform OOD detection.
> > As the results on SCOOD benchmark indicate, the learned $g(x)$ has made an effective step towards the balanced $g^{bal}(x)$, which translates into consistent improvements on **all 6 test subsets**.
> >
> > | SCOOD Test Set |  Method  |   AUROC↑  |    AUPR↑  |   FPR95↓  |
> > |:--------------:|:--------:|:---------:|:---------:|:---------:|
> > |     Texture    |   PASCL  |   93.16   |   84.80   |   23.26   |
> > |                | **Ours** | **96.22** | **93.50** | **17.60** |
> > |      SVHN      |   PASCL  |   96.63   |   98.06   |   12.18   |
> > |                | **Ours** | **97.12** | **98.08** | **10.47** |
> > |    CIFAR100    |   PASCL  |   84.43   |   82.99   |   57.27   |
> > |                | **Ours** | **86.03** | **85.67** | **50.84** |
> > |       TIN      |   PASCL  |   87.14   |   81.54   |   47.69   |
> > |                | **Ours** | **89.07** | **85.06** | **40.34** |
> > |      LSUN      |   PASCL  |   93.17   |   91.76   |   26.40   |
> > |                | **Ours** | **94.89** | **94.38** | **22.03** |
> > |    Places365   |   PASCL  |   91.43   |   96.28   |   33.40   |
> > |                | **Ours** | **93.06** | **97.19** | **28.30** |
> > |   **Average**  |   PASCL  |   90.99   |   89.24   |   33.36   |
> > |                | **Ours** | **92.73** | **92.31** | **28.27** |

---

> > > ### Author Response · Authors · 2023-11-19
> > > **Response to reviewer qdXa (Part. III)**
> > >
> > > > Q4: Ablation on the hyper-parameters and experimental settings.
> > >
> > > For a fair comparison, we follow PASCL's experimental settings in our manuscript and do not introduce extra manually-designed hyper-parameters.
> > > Here we report additional ablations on the choice of auxiliary OOD data and evaluation datasets respectively.
> > >
> > >
> > > 1. **Ablation on the choice of auxiliary OOD data.**
> > >
> > > In the main experiments reported in the manuscript, we follow PASCL to adopt the TinyImages80M $^{[5]}$ as auxiliary OOD data for training-time regularization, and the models are tested on the SCOOD benchmark for 6 subsets.
> > > As reviewer r7k6 suggests, we now train the models with CIFAR100 as auxiliary OOD data (CIFAR10-LT is ID), and test them on the remaining 5 subsets of the SCOOD benchmark (CIFAR100 was originally used as test OOD data but is now eliminated).
> > > According to the results reported below, our ImOOD consistently outperforms the advanced PASCL method.
> > >
> > > |  Method  |   AUROC↑  |    AUPR↑  |   FPR95↓  |
> > > |:--------:|:---------:|:---------:|:---------:|
> > > |   PASCL  |   88.05   |   86.63   |   38.69   |
> > > | **Ours** | **89.14** | **87.30** | **35.48** |
> > >
> > > However, models trained with CIFAR100 perform slightly worse than models trained with TinyImages80M, indicating that a more generally collected auxiliary OOD dataset also contributes to the final OOD detection performance, which is consistent with common sense.
> > > We will add the experiment and discussion.
> > >
> > > 2. **Ablation on the choice of evaluation datasets.**
> > >
> > > In the paper, we mainly compare different OOD detectors on the SCOOD benchmark, which is a composite evaluation dataset with 6 subsets covering different scenarios.
> > > Here we identify two representative subsets from SCOOD to further demonstrate our efficacy.
> > > Specifically, SVHN can be viewed as far OOD, and CIFAR100 can be seen as near OOD (with CIFAR10-LT as ID), as suggested by Fort et al $^{[6]}$.
> > > As we previously discussed, our ImOOD brings consistent enhancement against the strong baseline PASCL, especially on CIFAR100 (near OOD) test set with a decrease of 6.4% on FPR95.
> > >
> > > In addition, we also report the spurious OOD detection evaluation as suggested by Ming et al $^{[7]}$. The WaterBird ID dataset also suffers from the imbalance problem (on water birds and land birds), and Ming et al specifically collected a subset of Places as the purious OOD test set (with spurious correlation to background). The results below also demonstrate our method's robustness in dealing with spurious OOD problems.
> > >
> > > |     Evaluation Dataset    |  Method  |   AUROC↑  |    AUPR↑  |   FPR95↓  |
> > > |:-------------------------:|:--------:|:---------:|:---------:|:---------:|
> > > |      SCOOD Benchmark      |   PASCL  |   90.99   |   89.24   |   33.36   |
> > > |                           | **Ours** | **92.73** | **92.31** | **28.27** |
> > > |       SVHN (Far OOD)      |   PASCL  |   96.63   |   98.06   |   12.18   |
> > > |                           | **Ours** | **97.12** | **98.08** | **10.47** |
> > > |    CIFAR100 (Near OOD)    |   PASCL  |   84.43   |   82.99   |   57.27   |
> > > |                           | **Ours** | **86.03** | **85.67** | **50.84** |
> > > | WaterBird (Spurious OOD)  |   PASCL  |   89.59   |   91.05   |   33.73   |
> > > |                           | **Ours** | **90.63** | **92.49** | **30.11** |
> > >
> > > If there are other concerns that still bother you, please feel free to inform us and we are more than happy to provide further explanations.
> > >
> > >
> > > [1] Wang et al. Partial and Asymmetric Contrastive Learning for Out-of-Distribution Detection in Long-Tailed Recognition. ICML, 2022.
> > >
> > > [2] Menon et al. Long-tail Learning via Logit Adjustment. ICLR, 2021.
> > >
> > > [3] Kang et al. Decoupling representation and classifier for long-tailed recognition. ICLR, 2020.
> > >
> > > [4] Jiang et al. Detecting Out-of-distribution Data through In-distribution Class Prior. ICML, 2023.
> > >
> > > [5] Torralba et al. 80 million tiny images: A large data set for nonparametric object and scene recognition. TPAMI, 2008.
> > >
> > > [6] Fort et al. Exploring the Limits of Out-of-Distribution Detection. NeurIPS, 2021.
> > >
> > > [7] Ming et al. On the Impact of Spurious Correlation for Out-of-distribution Detection. AAAI, 2022.

---

### Author Response · Authors · 2023-11-19
**Thanks for reviewers' valuable time and insightful suggestions**

We thank all the reviewers for their time, insightful suggestions, and valuable comments. We are encouraged to see **ALL**  reviewers find our method **well-motivated** and **clearly presented**, with **grounded theoretical analysis** and **strong empirical performance** to demonstrate ImOOD's efficacy.

However, reviewers still have some extra concerns, which mainly focus on the following aspects:
* discussion and comparison with very recent related works;
* additional experiments and ablations on dataset choices;
* analysis and discussion on a few experimental phenomenons and method implementations.

We respond to each reviewer's comments in detail below, and here is a brief summary of the main discussions and experiments:
* supplementary discussion and comparison with related works, such as ClassPrior, OpenMax, OLTR, etc;
* extensive ablation studies on the choice of auxiliary OOD training data (e.g., synthesized OOD) and the evaluation on different OOD test data (e.g., near OOD and spurious OOD);
* detailed discussion and analysis on the estimation of probability distributions, as well as the effectiveness of the proposed post-hoc normalization technique.

We thank the reviewers' valuable suggestions again, and we believe those make our paper much stronger.

---

### Comment · Area_Chair_gmsX · 2023-11-21
**[Time Sensitive, ICLR24] Please read the authors' responses and try to discuss the remaining concerns with the authors**

Dear Reviewers,

The authors have provided detailed responses to your comments.

Could you have a look and try to discuss the remaining concerns with the authors? The reviewer-author discussion will end in two days.

We do hope the reviewer-author discussion can be effective in clarifying unnecessary misunderstandings between reviewers and the authors.

Best regards,

Your AC

---

### Meta-Review · Area_Chair_gmsX · 2023-12-06

**Metareview:**

The paper presents ImOOD, a novel statistical framework for Out-of-Distribution (OOD) detection in imbalanced data scenarios, which is a prevalent and challenging problem in real-world applications. The authors have commendably identified critical issues in current OOD detection methods and proposed a solution driven by Bayesian analysis, demonstrating the impacts of data imbalance. Their empirical demonstrations, such as those in Figure 1, and extensive ablation studies, notably in Table 4, showcase the strengths of ImOOD in addressing the limitations of existing techniques.

After the rebuttal, there is still a remaining issue that the difference between this work and previous work is not clear. The authors claim that they provide more theoretical support than Jiang's paper, however, it seems not. The math derived in this paper cannot be called rigorous theoretical support. there is no proper assumption analyzed in this paper. More importantly, the empirical support is discovered not enough after the rebuttal. Full experiments regarding the Imagenet LT should be included. RP+GradNorm outperforms the proposed method, but there is no comparison in the main paper. Thus, I recommend rejecting this paper at this round and encourage the authors to submit a good revision to the next machine learning conference.

**Justification For Why Not Higher Score:**

There is still a remaining issue that the difference between this work and previous work is not clear. More importantly, the empirical support is discovered not enough after the rebuttal. Full experiments regarding the Imagenet LT should be included. RP+GradNorm outperforms the proposed method, but there is no comparison in the main paper.

**Justification For Why Not Lower Score:**

N/A

---

### Decision · Program_Chairs · 2024-01-16

Reject